# Shapley Regularized Neural Granger Causality

**Maolin Yang** [1] [2]   **Zhoufan Zhu** [3] [4]   **Yuanhe Tian** [2]   **Kun Gao** [2] [*]   **Muyi Li** [3] [5] [6] [*]

## Abstract

Identifying temporal causal structure is fundamental to understanding complex systems. Neural Granger causality has emerged as a powerful paradigm for this task, leveraging the expressiveness of neural networks to model intricate nonlinear dynamics. Although complex architectures excel at predictive modeling, existing methods typically rely on simple local measures for causal discovery, which extract only partial information from the learned model and may miss global dependencies. To address this issue, we reformulate Granger causality as a feature attribution problem and propose the Information-Theoretic Shapley value (Info-Shap) to measure global feature importance. We first establish the theoretical equivalence between zero Info-Shap and Granger noncausality. On top of this, we construct two novel regularizers to suppress spurious relationships and mitigate overfitting. These regularizers are model-agnostic and can be seamlessly integrated into the training of any differentiable neural network. Through extensive experiments on synthetic and realistic datasets, we demonstrate that our method robustly recovers the underlying causal relationships, providing a flexible tool for causal discovery in high-dimensional nonlinear time series.

## 1. Introduction

Inferring causal structure from time series, known as temporal causal discovery, is the core objective in various scientific fields, ranging from economics and genomics to neuroscience ([Seth et al., 2015](); [Deshpande et al., 2022](); [Li et al., 2023]()). Despite its significance, reliable structural recovery in practice remains a challenge due to the presence of nonlinear dynamics, high dimensionality and data scarcity. In such regimes, traditional statistical tests ([Dhamala et al., 2008](); [Runge et al., 2019]()) often become computationally prohibitive, whereas classical linear vector autoregressive (VAR) models ([Lozano et al., 2009](); [Eichler, 2012]()) fail to capture nonlinear dependencies.

Recent advances in deep learning have introduced promising tools for temporal causal discovery, most notably by generalizing the classical notion of Granger causality ([Granger, 1969]()). This model-based paradigm, referred to as Neural Granger Causality (NGC; [Shojaie & Fox, 2022]()), leverages the expressive power of neural networks to infer causal relationships directly from the fitted model. By adopting specific neural architectures ([Cheng et al., 2023](); [Han et al., 2025](); [Poonia et al., 2025]()), NGC methods have demonstrated notable improvements over traditional approaches, capturing underlying dynamics with greater fidelity and scaling effectively to high-dimensional settings.

Despite these successes, a fundamental misalignment persists between *model expressiveness* and *inference mechanism*. While modern architectures excel at capturing complex nonlinear dynamics, the mechanism used to extract the causal graph often relies on heuristic proxies, such as weight magnitudes or local sensitivity analysis. These local measures are insufficient for deep models: they fail to account for variable interactions and saturation effects ([Shrikumar et al., 2017]()) inherent in non-linear functions. Consequently, a model may perfectly capture temporal dependencies, yet the inference mechanism fails to identify them, compromising the structural validity of the discovered graph.

To formalize this issue, we introduce a novel perspective that unifies a substantial portion of existing NGC methods. Grounded in *feature attribution*, our framework characterizes Granger causality through a structural triad: the importance measure, the predictive model, and the penalty term. As illustrated in Figure 1, we identify the importance measure as the cornerstone of the triad: it defines how the causal graph is inferred from the learned model, explicitly induces the regularization penalty, and implicitly constraint the architectural design of the predictive model.

---

[1]Paula and Gregory Chow Institute for Studies in Economics, Xiamen University, China [2]Zhongguancun Academy, Beijing, China [3]Wang Yanan Institute for Studies in Economics (WISE), Xiamen University, China [4]Department of Finance, School of Economics, Xiamen University, China [5]Department of Statistics and Data Science, School of Economics, Xiamen University, China [6]Key Laboratory of Econometrics (Xiamen University), Ministry of China, China. Correspondence to: Kun Gao <gaokun@bza.edu.cn>, Muyi Li <limuyi@xmu.edu.cn>.

*Proceedings of the 43rd International Conference on Machine Learning*, Seoul, South Korea. PMLR 306, 2026. Copyright 2026 by the author(s).

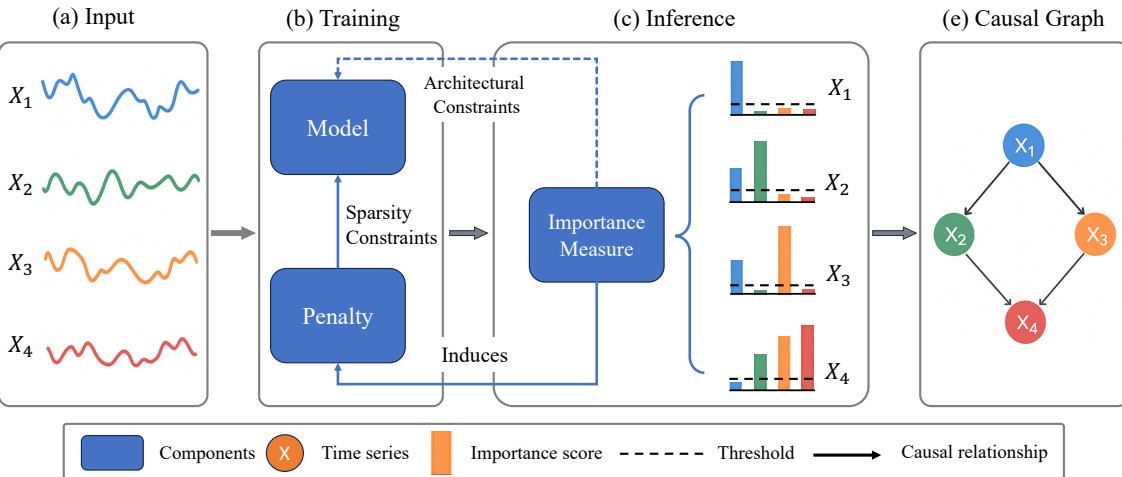

*Figure 1.* **Overview of the unified framework for model-based Granger causality**. The framework comprises three interdependent components: an importance measure, a predictive model, and a penalty term. *Crucially, the importance measure serves as the cornerstone of this triad.* It explicitly induces the penalty to promote sparsity (blue solid arrow) and, as our framework reveals, implicitly constrains the architecture of the model (blue dashed arrow). Despite its significance, this dependency has been largely overlooked in prior literature, which primarily favors complex architectures while neglecting the architectural constraints imposed by the choice of importance measure. We bridge this gap by introducing a principled, model-agnostic global importance measure that decouples inference from model architecture, enabling flexible model design without compromising inference fidelity.

Critically, while the predictive model and penalty term have received extensive attention (Khanna & Tan, 2020; Marcinkevičs & Vogt, 2021; Calvo-Pardo et al., 2021), the foundational role of the importance measure has been largely overlooked. As detailed in Appendix A, the majority of existing methods rely on local importance measures (Suryadi et al., 2023; Zhou et al., 2024). To overcome the limitations of these local proxies, the Shapley value (Shapley et al., 1953) offers a principled alternative. Although theoretically appealing for its axiomatic foundation (Lundberg & Lee, 2017; Covert et al., 2020), in practice, the resulting combinatorial complexity renders it computationally prohibitive for end-to-end training (Wang et al., 2021).

To bridge this gap, we propose the Information-Theoretic Shapley value (Info-Shap), a novel global feature importance measure grounded in information theory and cooperative game theory. By defining the coalition value function via generative information (Cheng & Tong, 2024), we characterize feature importance based on its information contribution to the output. Our approach leverages the axiomatic uniqueness of the Shapley value in attributing *marginal contribution*, which naturally aligns with the fundamental principle of Granger causality based on *incremental predictability*. Beyond this conceptual alignment, we further establish a formal equivalence between zero Info-Shap and Granger non-causality. Crucially, Info-Shap admits a closed-form expression, which explicitly decomposes both individual and interaction effects. This analytical tractability enables the construction of two theoretically grounded penalty terms: an exact derivation for fidelity and a scalable surrogate for

efficiency. These penalties are model-agnostic and can be incorporated into the training objectives of a broad class of network architectures. By aligning the structural inference mechanism with the model's expressiveness, our approach facilitates more reliable and interpretable temporal causal discovery in high-dimensional nonlinear systems.

We summarize our contributions as follows:

- We present a unified framework for model-based Granger causality, demonstrating that reliable causal discovery necessitates an inference mechanism commensurate with the model's expressiveness.

- We introduce Info-Shap, a global feature importance measure with a closed-form expression, and establish an equivalence between axiomatic feature attribution and Granger causality.

- We derive two model-agnostic regularization penalties that promote sparsity in feature importance and are compatible with a wide range of network architectures.

- We conduct extensive experiments on synthetic and realistic datasets, demonstrating that our method faithfully recovers the underlying causal structures.

## 2. Background and Related Work

### 2.1. Granger Causality

Adopting the general definition of Granger causality from Shojaie & Fox (2022), we consider a $d$-dimensional

weakly stationary time series $\{\mathbf{X}_t\}_{t=1}^T$, where $\mathbf{X}_t = (X_t^1, \ldots, X_t^d)^\top \in \mathbb{R}^d$. We denote the information set of series $i$ up to time $t-1$ by $\mathbf{X}_{<t}^i := (X_{t-1}^i, X_{t-2}^i, \ldots)^\top$, and the information set of all series by $\mathbf{X}_{<t} := (\mathbf{X}_{<t}^1, \ldots, \mathbf{X}_{<t}^d)^\top$. We assume the Data Generating Process (DGP) follows a nonlinear autoregressive process:

$$
\begin{aligned}
\mathbf{X}_t &= g(\mathbf{X}_{<t}) + \boldsymbol{\varepsilon}_t \\
&= \left[ g^1(\mathbf{X}_{<t}), \ldots, g^d(\mathbf{X}_{<t}) \right]^\top + \boldsymbol{\varepsilon}_t,
\end{aligned}
\tag{1}
$$

where the component function $g^i(\cdot)$ represents the temporal dependencies of the $i$-th series on the full information set, and $\boldsymbol{\varepsilon}_t \in \mathbb{R}^d$ is an independent noise vector with zero mean. Under this formulation, Granger causality is characterized by the dependence structure of the component functions:

**Definition 2.1.** Time series $j$ does not Granger-cause time series $i$ if and only if the $i$-th component function is invariant to the information set of the $j$-th series. That is, for all $t$, let $\mathbf{X}'_{<t} = (\mathbf{X}_{<t}^1, \ldots, \mathbf{X}_{<t}^{j\prime}, \ldots, \mathbf{X}_{<t}^d)^\top$ be a information set that differs from $\mathbf{X}_{<t}$ only in the $j$-th series. It holds that

$$
g^i(\mathbf{X}_{<t}) = g^i(\mathbf{X}'_{<t}).
\tag{2}
$$

Otherwise, $\{X_t^j\}$ is said to Granger-cause $\{X_t^i\}$.

This functional characterization allows us to parameterize $g(\cdot)$ using neural networks. The expressiveness of these models offers a practical avenue for causal discovery in high-dimensional, nonlinear scenarios.

## 2.2. Shapley Value

The Shapley value (Shapley et al., 1953) is a solution concept in cooperative game theory that provides an axiomatic framework to fairly distribute the total gains (or costs) among players. Particularly, consider a cooperative game with a set of players $N = \{1, 2, \ldots, n\}$ and a coalition value function $v : 2^N \to \mathbb{R}$, which assigns a real-valued payoff $v(S)$ to each coalition (subset) $S \subseteq N$. For player $i$, the Shapley value $\phi_i(v)$ is defined as:

$$
\phi_i(v) = \sum_{S \subseteq N \setminus \{i\}} \frac{|S|!(n - |S| - 1)!}{n!} [v(S \cup \{i\}) - v(S)].
$$

Conceptually, $\phi_i(v)$ represents the *expected marginal contribution* of player $i$ when joining an unknown coalition $S \subseteq N \setminus \{i\}$. By aggregating contributions across all possible $S$, it serves as a global measure of player $i$'s importance to the game. Beyond its intuitive appeal, the Shapley value is the unique solution satisfying the axioms of efficiency, symmetry, additivity and dummy (Winter, 2002). In our framework, the dummy axiom is particularly fundamental (see Theorem 3.2), as it establishes a direct linkage between feature importance and Granger causality.

## 2.3. Related Work

**Granger Causality.** Introduced half a century ago, Granger causality (Granger, 1969) has become a popular framework for temporal causal discovery. Early approaches primarily focus on bivariate settings, utilizing statistical tests to assess the significance of predictive improvements (Chamberlain, 1982; Geweke, 1982; Dhamala et al., 2008). Subsequent work generalized this to multivariate time series using VAR models (Eichler, 2012), and various regularization techniques have been proposed to mitigate the curse-of-dimensionality (Fujita et al., 2007; Lozano et al., 2009; Nicholson et al., 2020). Recently, there has been a shift toward NGC (Shojaie & Fox, 2022), which leverages the modeling capacity of neural networks to capture nonlinear dynamics. As highlighted by Tank et al. (2021), the black-box nature of neural networks poses challenges for causal interpretation. To address this, existing NGC methods mainly follow two strategies: (i) constrained joint modeling, which imposes specific structural forms on $g(\cdot)$ to achieve interpretability (Marcinkevičs & Vogt, 2021; Bussmann et al., 2021; Han et al., 2025); or (ii) component-wise modeling, which fits separate models for each $g^i(\cdot)$ to disentangle the effects of input variables (Khanna & Tan, 2020; Tank et al., 2021; Poonia et al., 2025). A notable exception is Zhou et al. (2024), which employs the input-output Jacobian as a proxy for Granger causality, marking the first work to bypass architectural constraints. However, viewed through the lens of feature attribution, this approach remains limited to local sensitivity. In contrast, our method leverages the axiomatic foundations of the Shapley value to measure global feature importance, thereby resolving the misalignment between the model's expressiveness and the inference mechanism.

**Feature Attribution.** Classical approaches quantify feature importance by analyzing the statistical or geometric properties of the response surface. Prominent examples include variance-based Sobol' indices (Sobol, 2001; Fel et al., 2021), derivative-based measures (Sobol & Kucherenko, 2010), and active subspace methods (Constantine & Diaz, 2017). To provide a unified axiomatic foundation (Covert et al., 2021), recent work has increasingly adopted the Shapley value (Shapley et al., 1953) from cooperative game theory. Within this paradigm, various methods arise from distinct specifications of the coalition value functions (Song et al., 2016; Lundberg & Lee, 2017; Covert et al., 2020). While these methods provide valuable post-hoc insights, such information cannot be leveraged during model training, because the exact computation of the Shapley value is prohibitive in high-dimensional settings (Wang et al., 2021). By comparison, our Info-Shap admits a closed-form expression. This analytical tractability alleviates the computational bottleneck, allowing the incorporation of attribution information into the training objective.

# 3. Methodology

## 3.1. Generative Information in Supervised Learning

Consider a supervised learning setting with an input vector $\mathbf{X} = (X^1, \ldots, X^d) \in \mathbb{R}^d$ and a univariate target $Y \in \mathbb{R}$. To construct an information-theoretic value function for feature coalitions, we extend the concept of generative information (Cheng & Tong, 2024) to the conditional distribution $p(Y \mid \mathbf{X})$. Specifically, we define the Conditional Generative Information Matrix (CGIM) as:

$$\mathrm{CGIM}_{Y|\mathbf{X}} = \mathbb{E}_{Y|\mathbf{X}}[\nabla_{\mathbf{X}} \log p(Y|\mathbf{X}) \nabla_{\mathbf{X}} \log p(Y|\mathbf{X})^\top],$$

where $\nabla_{\mathbf{X}} \log p(Y|\mathbf{X})$ is a gradient vector. By defining a Riemannian metric on the input manifold, the CGIM characterizes the information geometry of the input space by measuring the sensitivity of the target distribution to input perturbations. This geometric interpretation quantifies the information flow from $\mathbf{X}$ to $Y$, providing a natural basis for evaluating the value of feature coalitions within the cooperative game.

To obtain an analytically tractable value function, we consider the Gaussian assumption on conditional distribution $Y \mid \mathbf{X} \sim \mathcal{N}(g(\mathbf{X}), \sigma^2)$, where $g : \mathbb{R}^d \to \mathbb{R}$ is a differentiable function representing the structural relationship between $\mathbf{X}$ and $Y$. Under this specification, the CGIM simplifies to (see Appendix B.1 for derivations):

$$\mathrm{CGIM}_{Y|\mathbf{X}} = \frac{1}{\sigma^2} \left( \partial g \partial g^\top \right), \qquad (3)$$

where $\partial g = \partial g / \partial \mathbf{X}$ denotes the gradient vector. This formulation explicitly links the curvature of the log-likelihood to the variations in the predictive mean, effectively bridging feature attribution with information-theoretic dependence.

We impose this Gaussian assumption primarily for analytical expedience. As shown in Appendix B.3, the closed-form expression for the Shapley value (5) follows from the 2-additive structure (Grabisch, 1997) of the value function (4), rather than from Gaussianity per se. This observation suggests that analogous closed-form results may be possible without the Gaussian specification, provided that the CGIM-induced value function remains 2-additive. In such settings, however, the main challenge lies in estimating the score function and the resulting Shapley value, which are technically more involved and beyond the scope of the present work.

## 3.2. Information-Theoretic Shapley Value

Building on the CGIM formulation in (3), we now define the information contribution of a feature coalition $\mathbf{X}^S = \{X^i\}_{i \in S}$ to the target $g(\mathbf{X})$. While we restrict the main text to scalar targets to demonstrate the methodology, the general multivariate framework is established in

Appendix B.2, serving as the foundation for the Granger causality analysis in Section 3.4. All technical proofs are deferred to Appendix B.

Rigorously, we consider a cooperative game $(\mathcal{N}, v)$ where the set of players $\mathcal{N} = \{1, \ldots, d\}$ correspond to the input features. We define the coalition value function as:

$$v_{\mathrm{Info}}(S) = \mathbb{E}_{\mathbf{X}^S} \left( \sum_{i,j \in S, i \leq j} |\partial_i g \, \partial_j g| \right), \qquad (4)$$

where $\partial_i g = \partial g(\mathbf{X}) / \partial X^i$ denotes the partial derivative of $g(\cdot)$ with respect to $X^i$, and the expectation is taken over the joint distribution of $\mathbf{X}^S$. Essentially, $v_{\mathrm{Info}}(S)$ captures the predictive influence of $\mathbf{X}^S$ by aggregating the expected magnitudes of all unique entries within the corresponding CGIM sub-matrix indexed by $S$.

One notable advantage of $v_{\mathrm{Info}}(S)$ is that it admits a closed-form expression for the Shapley value, as stated in the following theorem.

**Theorem 3.1** (Information-Theoretic Shapley Value). *The Shapley value of feature $i$ with respect to value function $v_{Info}(S)$ is given by*

$$\phi_i(v_{Info}) = \underbrace{\mathbb{E}\left(|\partial_i g|^2\right)}_{\text{individual effect}} + \underbrace{\frac{1}{2} \sum_{j \neq i} \mathbb{E}(|\partial_i g \, \partial_j g|)}_{\text{interaction effect}}. \qquad (5)$$

We designate (5) as Info-Shap to emphasize its information-theoretic foundation.

Theorem 3.1 reveals that Info-Shap measures global feature importance through a natural decomposition into two terms. The first characterizes the marginal contribution of feature $i$, while the second accounts for the pairwise interaction between feature $i$ and the remaining features. Although Duan & Okten (2025) presents a mathematically similar expression motivated by activity score (Constantine & Diaz, 2017), their formulation lacks a clear game-theoretic interpretation.

In contrast, Info-Shap is derived from a well-defined cooperative game $(\mathcal{N}, v_{\mathrm{Info}})$ specifically formulated to quantify information contribution. This principled construction is pivotal, as it explicitly connects the game-theoretic concept of 'value' to the statistical notion of 'predictive information' required for identifying Granger causality. As formalized below, the dummy axiom establishes an equivalence between feature attribution and conditional mean independence:

**Theorem 3.2** (Dummy Axiom). *The Info-Shap of feature $X^i$ is zero, i.e., $\phi_i(v_{Info}) = 0$, if and only if $X^i$ is conditionally mean independent of $Y$ given all of the remaining features $\mathbf{X}^{-i} = [X^1, \ldots, X^{i-1}, X^{i+1}, \ldots, X^d]^\top$. That is,*

$$\mathbb{E}[Y \mid X^i, \mathbf{X}^{-i}] = \mathbb{E}[Y \mid \mathbf{X}^{-i}]. \qquad (6)$$

Theorem 3.2 provides a principled foundation for identifying Granger causality. According to Definition 2.1, a series $\{X_t \in \mathbb{R}\}$ does not Granger-cause $\{Y_t \in \mathbb{R}\}$ if its historical observations provide no additional predictive information regarding $Y_t$. That is, the conditional expectation of $Y_t$ is invariant to the past values of $\{X_t\}$:

$$g(Y_{<t}, X_{<t}) = g(Y_{<t})$$
$$\iff \quad \mathbb{E}[Y_t \mid Y_{<t}, X_{<t}] = \mathbb{E}[Y_t \mid Y_{<t}]. \quad (7)$$

Under supervised learning, this invariance necessitates that the Info-Shap of all associated lagged features of $\{X_t\}$ are zero. Consequently, we can reformulate Granger causality identification as a feature attribution problem, where a causal relationship is identified by evaluating whether the Info-Shap of the corresponding lagged features is non-zero.

### 3.3. Shapley Regularizer

Theorem 3.2 establishes the theoretical foundation for identifying Granger causality using Info-Shap. In practice, this requires estimating the unknown mechanism $g(\cdot)$ from finite data. Without appropriate structural constraints, overparameterized neural networks are prone to fit spurious correlations, leading to dense graphs that contradict the inherent sparsity of real-world systems (Zhou et al., 2024). To address this, we introduce two regularization penalties induced by Info-Shap, which act as inductive biases to encourage sparse feature attributions and mitigate overfitting.

We first introduce the *Shapley regularizer* (Shap), which imposes an $\ell_1$ penalty directly on the Info-Shap values of all features:

$$\Omega_{\text{Shap}}(g; \lambda) = \lambda \sum_{i=1}^{d} |\phi_i(v_{\text{Info}})| \quad (8)$$

where $\lambda > 0$ is the regularization hyperparameter. By constraining the overall magnitude of feature attribution, $\Omega_{\text{Shap}}(g; \lambda)$ encourages the model to focus on the most informative features. By exploiting automatic differentiation in modern deep learning frameworks, we can easily compute the required gradients during training, allowing the Shap to be directly integrated into the training objectives of most network architectures.

The exact computation of Shap necessitates materializing all derivatives, which can be computationally expensive in high-dimensional settings (Hoffman et al., 2019). To facilitate efficient computation, we first observe that $\Omega_{\text{Shap}}(g; \lambda)$ admits a decomposition into individual and interaction components (derivation provided in Appendix B.1):

$$\Omega_{\text{Shap}}(g; \lambda) = \lambda \left[ \frac{1}{2} \mathbb{E} \|\partial g\|_2^2 + \frac{1}{2} \mathbb{E} \|\partial g \partial g^T\|_{\ell_1} \right], \quad (9)$$

where $\|\cdot\|_2$ denotes the Euclidean norm and $\|\cdot\|_{\ell_1}$ denotes the element-wise $\ell_1$ norm. The computational bottleneck

arises specifically from the interaction term: the non-smooth $\ell_1$ norm precludes efficient implicit evaluation, requiring the full interaction matrix $\partial g \partial g^T$ to be instantiated.

To overcome this scalability barrier, we propose the *Fast-Shapley Regularizer* (F-Shap). This formulation relaxes the element-wise $\ell_1$ norm on interaction term to a squared Frobenius norm $\|\cdot\|_F^2$, a crucial modification that enables efficient stochastic approximation via random projections. Analytically, we define the F-Shap as:

$$\Omega_{\text{F-Shap}}(g; \lambda_1, \lambda_2) = \frac{\lambda_1}{2} \mathbb{E} \|\partial g\|_2^2 + \frac{\lambda_2}{2} \mathbb{E} \|\partial g \partial g^T\|_F^2, \quad (10)$$

where $\lambda_1$ and $\lambda_2$ control the regularization strength for the individual and interaction effects. Notably, the Jacobian regularizer (Hoffman et al., 2019; Zhou et al., 2024) corresponds to the special case of F-Shap when the interaction penalty is removed (i.e., $\lambda_2 = 0$).

We avoid explicit matrix materialization by employing Hutchinson's trace estimator (Hutchinson, 1989) to approximate the squared Frobenius norm, reducing the computational complexity from linear to constant in the output dimension. The detailed approximation scheme and the corresponding theoretical complexity analysis are provided in Appendix C.1, while empirical verification of the method's accuracy and efficiency is presented in Appendix D.

### 3.4. Shapley Regularized Neural Granger Causality

In this section, we extend the proposed Info-Shap and the induced penalties to multivariate time series for temporal causal discovery. We term this approach **S**hapley **R**egularized **N**eural **G**ranger **C**ausality, using SRNGC and F-SRNGC to denote the variants that employ the exact calculation of Shap and efficient approximation of F-Shap as regularizers, respectively.

**Model Formulation.** Assume $\{\mathbf{X}_t\}_{t=1}^{T}$ is generated by a nonlinear autoregressive process of order $K$. Accordingly, we construct the lagged feature vector as $\widetilde{\mathbf{X}}_{t-1} = [\mathbf{X}_{t-1}^\top, \ldots, \mathbf{X}_{t-K}^\top]^\top \in \mathbb{R}^{Kd}$. To model the temporal dependence, we parameterize the conditional expectation using a neural network $g_\theta(\cdot): \mathbb{R}^{Kd} \to \mathbb{R}^d$. The corresponding one-step-ahead prediction at time $t$ is given by $\widehat{\mathbf{X}}_t = g_\theta(\widetilde{\mathbf{X}}_{t-1})$. For notational compactness, we let $T' = T - K$ represent the effective sample size and use $\sum_t$ to denote the summation over the valid time indices $t = K + 1, \ldots, T$.

**Training Objective.** We train the neural network $g_\theta(\cdot)$ by minimizing a composite objective that balances prediction accuracy with structural sparsity. Specifically, the objective function is defined as:

$$\mathcal{L}(\theta) = \frac{1}{T'} \sum_t \|\mathbf{X}_t - g_\theta(\widetilde{\mathbf{X}}_{t-1})\|_2^2 + \widehat{\Omega}(g_\theta; \lambda), \quad (11)$$

where the first term represents the mean squared prediction error, and $\widehat{\Omega}(g_\theta; \lambda)$ denotes the empirical Shapley regularizer with penalty strength $\lambda$, as defined below.

**Empirical Regularizer.** The calculation of Shap (9) and F-Shap (10) involves expectations over the input distribution, which we estimate using empirical averages over training data. To this end, we define the local output-input Jacobian matrix at time $t$ as $\mathcal{J}_t = \partial g_\theta(\widetilde{\mathbf{X}}_{t-1})/\partial \widetilde{\mathbf{X}}_{t-1} \in \mathbb{R}^{d \times Kd}$, where the $i$-th row $\mathcal{J}_{t,i} \in \mathbb{R}^{1 \times (Kd)}$ represents the gradient vector of $i$-th output. Using this notation, the proposed Shapley regularizers are computed as:

$$\widehat{\Omega}_{\text{Shap}}(g_\theta; \lambda) = \frac{\lambda}{T'} \sum_t \left( \|\mathcal{J}_t\|_F^2 + \sum_{i=1}^d \|\mathcal{J}_{t,i}^\top \mathcal{J}_{t,i}\|_{\ell_1} \right). \quad (12)$$

The F-Shap (10) is approximated similarly using $\mathcal{J}_t$ and the algorithm described in Appendix C.1.

**Feature Attribution.** To infer the causal graph from the learned model $\hat{g}(\cdot)$, we quantify the global importance of each lagged feature using Info-Shap. Let $\mathcal{J}_{t,j,k}^i = \partial \hat{g}^i(\widetilde{\mathbf{X}}_{t-1})/\partial X_{t-k}^j$ denote the partial derivative of the $i$-th output series with respect to the $j$-th input series at lag $k$ and time $t$. We calculate the Info-Shap as follows:

$$\widehat{\phi}_{ijk} = \frac{1}{T'} \sum_t \left( |\mathcal{J}_{t,j,k}^i|^2 + \frac{1}{2} \sum_{j',k'} |\mathcal{J}_{t,j,k}^i \mathcal{J}_{t,j'k'}^i| \right), \quad (13)$$

where the inner sum runs over all $(j', k') \neq (j, k)$. The empirical Info-Shap $\hat{\phi}_{ijk}$ quantifies the marginal information contribution of the $j$-th series at lag $k$ to the prediction of the $i$-th series. By Theorem 3.2, if $\hat{\phi}_{ijk} = 0$ for all $k$ from 1 to $K$, then series $j$ does not Granger-cause series $i$. Consequently, evaluating the Info-Shap for all pairs allows us to reconstruct the causal graph.

**Knockoff-style Thresholding.** In practice, empirical estimates of Info-Shap may be nonzero even for irrelevant features due to finite-sample noise and approximation error. To separate strong attributions from likely spurious ones, we employ a data-adaptive thresholding procedure inspired by the knockoff filter (Barber & Candès, 2015).

We first construct synthetic knockoff features $\widetilde{\mathbf{X}}_{t-1}^{\text{ko}}$ by residual bootstrap. Specifically, we define:

$$\widetilde{\mathbf{X}}_{t-1}^{\text{ko}} = \hat{g}(\widetilde{\mathbf{X}}_{t-1}) + \hat{\varepsilon}_t^*, \quad (14)$$

where $\hat{\varepsilon}_t = \mathbf{X}_t - \hat{g}(\widetilde{\mathbf{X}}_{t-1})$ are the fitted residuals, and $\hat{\varepsilon}_t^*$ are bootstrap samples drawn from $\{\hat{\varepsilon}_t\}$. We then retrain the model on the augmented feature set $[\widetilde{\mathbf{X}}_{t-1}^\top, \widetilde{\mathbf{X}}_{t-1}^{\text{ko}\top}]^\top$ and compute the Info-Shap for both original and knockoff features, denoted by $\hat{\phi}_{ijk}^*$ and $\hat{\phi}_{ijk}^{\text{ko}}$, respectively. Next, we

*Table 1.* Performance comparison on the VAR(3) dataset. Mean AUROC and AUPRC scores are averaged over five independent realizations (standard deviations are indicated in parentheses). Best results are highlighted in **bold**.

| Metric | Penalty | $d = 10$ | | $d = 50$ | |
|--------|---------|----------|----------|----------|----------|
| | | $T = 500$ | $T = 1000$ | $T = 500$ | $T = 1000$ |
| AUROC | Jacob-$\ell_1$ | 0.999 (0.001) | **1.000** (0.000) | 0.895 (0.008) | **0.979** (0.003) |
| | Jacob-F | 0.998 (0.003) | **1.000** (0.000) | 0.877 (0.011) | 0.955 (0.003) |
| | Shap | 0.999 (0.002) | **1.000** (0.000) | **0.898** (0.009) | 0.977 (0.005) |
| | F-Shap | **1.000** (0.000) | **1.000** (0.000) | 0.882 (0.011) | 0.958 (0.003) |
| AUPRC | Jacob-$\ell_1$ | 0.998 (0.003) | 1.000 (0.001) | 0.855 (0.009) | **0.970** (0.004) |
| | Jacob-F | 0.996 (0.005) | 1.000 (0.001) | 0.831 (0.012) | 0.936 (0.004) |
| | Shap | 0.998 (0.004) | **1.000** (0.000) | **0.865** (0.008) | 0.970 (0.005) |
| | F-Shap | **1.000** (0.001) | **1.000** (0.000) | 0.835 (0.009) | 0.940 (0.006) |

calculate the knockoff-style test statistic as:

$$W_{ijk} = \widehat{\phi}_{ijk}^* - \widehat{\phi}_{ijk}^{\text{ko}}. \quad (15)$$

Intuitively, a large positive value of $W_{ijk}$ indicates that the original feature has a stronger attribution than its knockoff counterpart, providing evidence for a causal effect from series $j$ to series $i$ at lag $k$. Following the knockoff framework, we select a data-adaptive threshold

$$\alpha_q = \min \left\{ \alpha > 0 : \frac{|\{(i,j,k) : W_{ijk} < -\alpha\}|}{|\{(i,j,k) : W_{ijk} > \alpha\}| \vee 1} \leq q \right\},$$

where $|\{\cdot\}|$ denotes the cardinality of the set, and $q$ serves as a nominal calibration parameter analogous to a target False Discovery Rate (FDR). The final causal graph is constructed by including all edges $(j \rightarrow i)$ where $W_{ijk} \geq \alpha_q$ for any lag $k$.

Under time series dependence, the residual-bootstrap knockoffs may not provide theoretical FDR guarantees. Nevertheless, the empirical results in Section 4.3 and Appendix D demonstrate that this data-driven thresholding procedure effectively helps suppress false discoveries.

## 4. Experiments

We empirically evaluate the proposed method on both synthetic and realistic datasets. To ensure fair comparison with baselines, we report the Area Under the Receiver Operating Characteristic (AUROC) and Area Under the Precision-Recall (AUPRC) curves using the raw importance scores prior to knockoff thresholding. Specifically, we define the $(i, j)$-th entry of the causal graph as the maximum score over all lags, i.e., $\max_k \hat{\phi}_{ijk}$. Besides these metrics, we also

*Table 2.* Performance comparison on the Lorenz-96 dataset.

| Metric | Penalty | $F = 10$ | | $F = 40$ | |
|---|---|---|---|---|---|
| | | $T = 500$ | $T = 1000$ | $T = 500$ | $T = 1000$ |
| AUROC | Jacob-$\ell_1$ | 0.694 (0.007) | 0.774 (0.031) | **0.746** (0.032) | 0.941 (0.122) |
| | Jacob-F | 0.652 (0.015) | 0.686 (0.014) | 0.727 (0.035) | 0.747 (0.024) |
| | Shap | **0.717** (0.011) | **0.802** (0.010) | 0.745 (0.022) | **0.984** (0.015) |
| | F-Shap | 0.667 (0.018) | 0.697 (0.017) | 0.726 (0.017) | 0.775 (0.025) |
| AUPRC | Jacob-$\ell_1$ | 0.428 (0.009) | 0.551 (0.043) | **0.500** (0.041) | 0.889 (0.217) |
| | Jacob-F | 0.363 (0.009) | 0.394 (0.005) | 0.444 (0.028) | 0.477 (0.028) |
| | Shap | **0.439** (0.016) | **0.555** (0.015) | 0.489 (0.035) | **0.956** (0.023) |
| | F-Shap | 0.374 (0.013) | 0.394 (0.011) | 0.436 (0.025) | 0.495 (0.032) |

*Table 3.* AUROC performance on the DREAM3 dataset.

| Methods | Ecoli-1 | Ecoli-2 | Yeast-1 | Yeast-2 | Yeast-3 |
|---|---|---|---|---|---|
| Rhino | 0.685 | 0.680 | 0.664 | 0.585 | 0.588 |
| N.NTS | 0.567 | 0.515 | 0.571 | 0.512 | 0.524 |
| cMLP | 0.644 | 0.568 | 0.585 | 0.506 | 0.528 |
| TCDF | 0.614 | 0.647 | 0.581 | 0.556 | 0.557 |
| eSRU | 0.660 | 0.629 | 0.627 | 0.557 | 0.550 |
| GVAR | 0.652 | 0.634 | 0.623 | 0.570 | 0.554 |
| CUTS+ | 0.662 | 0.571 | 0.637 | 0.585 | 0.586 |
| JRNGC | 0.666 | 0.678 | 0.650 | 0.597 | 0.560 |
| SRNGC | **0.700** | **0.702** | **0.665** | **0.611** | 0.596 |
| F-SRNGC | 0.632 | 0.642 | 0.665 | 0.604 | **0.611** |

*Note:* PCMCI and LCCM are excluded from the DREAM3 experiments due to excessive computational costs.

present the binary causal graph inferred via knockoff thresholding in Section 4.3. In accordance with Zhou et al. (2024), we employ a Residual MLP (Das et al., 2024) to model the underlying temporal dynamics for all experiments. Detailed dataset descriptions are provided in Appendix D.1. The code and data required to reproduce all experiments are available at `https://github.com/MerlynYang/SRNGC`.

## 4.1. Synthetic Experiments

In this section, we benchmark the proposed *global* Info-Shap measure against *local* Jacobian-based baselines (Zhou et al., 2024) using two synthetic datasets. Since the causal structures and data-generating mechanisms are known, synthetic data allow us to directly assess how the choice of importance measure affects the fidelity of the inferred causal graphs. To systematically compare these measures in practice, we evaluate each under two regularization schemes: the element-wise $\ell_1$ norm and the Frobenius-norm variant approximated by a single random projection. Together, these choices yield four comparative penalties: Jacob-$\ell_1$, Jacob-F, Shap, and F-Shap.

*Table 4.* Performance comparison on CausalTime datasets. Mean AUROC and AUPRC are reported across 5 independent runs.

| Methods | AUROC | | | AUPRC | | |
|---|---|---|---|---|---|---|
| | AQI | Traffic | Medical | AQI | Traffic | Medical |
| PCMCI | 0.527 (0.074) | 0.542 (0.074) | 0.699 (0.011) | 0.673 (0.037) | 0.347 (0.058) | 0.508 (0.018) |
| LCCM | 0.857 (0.065) | 0.555 (0.025) | 0.801 (0.022) | 0.926 (0.025) | 0.591 (0.048) | **0.755** (0.024) |
| Rhino | 0.670 (0.098) | 0.627 (0.019) | 0.652 (0.021) | 0.759 (0.076) | 0.377 (0.009) | 0.490 (0.032) |
| N.NTS | 0.573 (0.023) | 0.633 (0.034) | 0.502 (0.068) | 0.710 (0.023) | 0.577 (0.054) | 0.457 (0.016) |
| cMLP | 0.717 (0.008) | 0.603 (0.006) | 0.574 (0.010) | 0.718 (0.007) | 0.358 (0.050) | 0.464 (0.012) |
| TCDF | 0.415 (0.021) | 0.503 (0.004) | 0.633 (0.038) | 0.653 (0.009) | 0.364 (0.005) | 0.554 (0.031) |
| eSRU | 0.823 (0.032) | 0.599 (0.019) | 0.756 (0.037) | 0.722 (0.032) | 0.489 (0.034) | 0.735 (0.060) |
| GVAR | 0.623 (0.041) | 0.633 (0.005) | 0.713 (0.019) | 0.790 (0.018) | 0.585 (0.002) | 0.677 (0.036) |
| CUTS+ | 0.893 (0.021) | 0.618 (0.075) | **0.820** (0.017) | 0.798 (0.088) | 0.637 (0.120) | 0.548 (0.135) |
| JRNGC | 0.928 (0.001) | 0.729 (0.005) | 0.754 (0.004) | 0.783 (0.002) | 0.594 (0.007) | 0.726 (0.002) |
| SRNGC | 0.946 (0.007) | 0.795 (0.017) | 0.720 (0.015) | 0.900 (0.009) | 0.622 (0.010) | 0.685 (0.011) |
| F-SRNGC | **0.961** (0.004) | **0.820** (0.012) | 0.736 (0.014) | **0.929** (0.007) | **0.682** (0.015) | 0.707 (0.011) |

Particularly, we employ two widely established simulated datasets: the linear VAR(3) model (Shojaie & Fox, 2022) and the nonlinear Lorenz-96 system (Lorenz, 1996) to illustrate the application of different methods. We consider dimensions $d \in \{10, 50\}$ for VAR(3), while fixing $d = 50$ for Lorenz-96. In the latter case, we further vary the forcing parameter $F \in \{10, 40\}$ to modulate the system's chaotic behavior. For each configuration, we generate 5 realizations with sample size $T \in \{500, 1000\}$, following the DGP detailed in Appendix D.1. Tables 1 and 2 present the results for VAR(3) and Lorenz-96, respectively.

We summarize the main observations as follows. First, while Frobenius-norm variants achieve computational efficiency via approximation, they entails a noticeable performance cost relative to the exact $\ell_1$ norm penalties; this trade-off aligns with findings in Zhou et al. (2024). Second, for VAR(3) model, four methods achieve comparable performance. This result aligns with theoretical expectations, as the Jacobian matrix full characterizes the linear relationships. Third, for Lorenz-96 model, Shapley-based methods generally outperform the respective Jacobian baselines within the same regularization scheme. This demonstrates that global measures are essential for capturing complex dependencies that local sensitivity methods overlook.

## 4.2. Realistic Benchmarks

In this section, we evaluate our method on two realistic benchmark datasets: DREAM3 (Prill et al., 2010) and

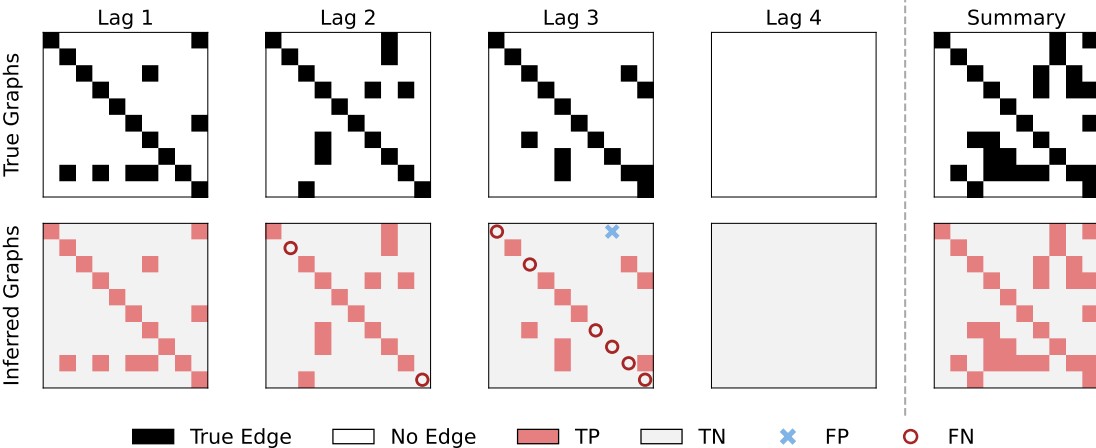

*Figure 2.* **Visualization of inferred Granger causal graphs.** We compare the ground truth (upper panel) against the SRNGC inferred graph (lower panel) on one realization of VAR(3) model ($d = 10, T = 1000$) with target FDR level set to be $q = 0.01$. The vertical divider separates the lagged graphs (left) from the final summary graphs (right). In the ground truth, black cells represent true edges (i.e., $\phi_{ijk} \neq 0$), where white cells represent the absence of an edge. In the estimated panels, performance is colored: red cells indicate true positives, gray cells indicate true negatives, blue crosses ($\times$) denote false positives, and red circles ($\circ$) denote false negatives.

CausalTime (Cheng et al., 2024b). These datasets are generated to mimic real-world scenarios while providing the true causal graphs, enabling quantitative validation of temporal causal discovery methods. Together, they represent two complementary settings commonly encountered in practice.

The DREAM3 dataset corresponds to a high-dimensional setting with limited sample size. It consists of five gene regulatory networks, each containing the expression levels of $d = 100$ genes. Specifically, each network comprises 46 independent replicates, and each replicate is sampled at 21 time points. In contrast, the CausalTime dataset represents a moderate-dimensional setting with sufficient sample size. This benchmark includes three realistic datasets: Traffic ($d = 20$), Medical ($d = 20$), and AQI ($d = 36$), each consisting of 480 replicates sampled at 40 time points.

We compare our method against two broad categories of baselines: (1) **NGC-based methods**, spanning component-wise modeling (cMLP (Tank et al., 2021), TCDF (Nauta et al., 2019), eSRU (Khanna & Tan, 2020)) and joint modeling (GVAR (Marcinkevičs & Vogt, 2021), CUTS+ (Cheng et al., 2024a), JRNGC (Zhou et al., 2024)); and (2) **general temporal causal discovery approaches**, including PCMCI (Runge et al., 2019), LCCM (De Brouwer et al., 2020), N.NTS (Sun et al., 2023), and Rhino (Gong et al., 2023).

We summarize the results in Tables 3 and 4. Overall, our empirical results show that the proposed method consistently outperforms competing baselines across most datasets. On the DREAM3 dataset, the observed improvements align with the findings of Cui et al. (2022), providing further evidence that Info-Shap effectively captures the interactions between features inherent in gene regulatory networks.

Furthermore, on the CausalTime benchmarks, our method achieves state-of-the-art performance on both AQI and Traffic datasets, demonstrating its robustness across diverse real-world scenarios. In contrast, performance on the Medical dataset is relatively modest, with our approach underperforming the component-wise eSRU baseline. We attribute this specific outcome to the inadequacy of the underlying predictive model, rather than a failure of the proposed global importance measure. This observation underscores the importance of network architectural design, as the efficacy of any causal discovery mechanism is naturally bounded by the expressiveness of the predictive model employed.

We also observe that the relative performance between SRNGC and F-SRNGC exhibits a clear dependence on the sample size. When data are limited (e.g., DREAM3), the approximation error of random projections is non-negligible, leading to SRNGC outperforming F-SRNGC. Conversely, for the CausalTime benchmarks, this error diminishes as sample size increases (Hoffman et al., 2019). In this context, F-SRNGC gains greater flexibility by separately penalizing individual and interaction effects, resulting in better performance. This finding offers a practical criterion for selecting between the two variants based on data availability.

### 4.3. Inferred Causal Graph

To demonstrate the reliability of the knockoff thresholding procedure for recovering binary causal graphs, we applied SRNGC to a representative realization of the VAR(3) model with $d = 10$ and $T = 1000$. We intentionally overspecified the model lag order to 4 (exceeding the true lag of 3) to evaluate the method's ability to suppress spurious causal

*Table 5.* Performance comparison on fMRI datasets. Results are averaged over the first 5 subjects (standard deviations in parentheses). Best results are highlighted in **bold**.

| Model | AUROC | | | | | AUPRC | | | | |
|---|---|---|---|---|---|---|---|---|---|---|
| | Layer-weight | Jacob-$\ell_1$ | Jacob-F | Shap | F-Shap | Layer-weight | Jacob-$\ell_1$ | Jacob-F | Shap | F-Shap |
| MLP | – | 0.859 (0.026) | 0.844 (0.030) | **0.874** (0.015) | 0.855 (0.025) | – | 0.645 (0.038) | 0.640 (0.036) | **0.673** (0.021) | 0.659 (0.029) |
| cMLP | 0.726 (0.016) | 0.827 (0.023) | 0.831 (0.018) | **0.854** (0.012) | 0.844 (0.018) | 0.429 (0.042) | 0.630 (0.020) | 0.633 (0.015) | **0.660** (0.014) | 0.639 (0.021) |
| LSTM | – | 0.836 (0.013) | 0.737 (0.049) | **0.847** (0.003) | 0.713 (0.051) | – | 0.590 (0.039) | 0.408 (0.071) | **0.672** (0.030) | 0.370 (0.076) |
| cLSTM | 0.681 (0.026) | 0.754 (0.051) | 0.705 (0.043) | **0.797** (0.049) | 0.681 (0.057) | 0.288 (0.015) | 0.491 (0.061) | 0.388 (0.032) | **0.567** (0.059) | 0.343 (0.065) |

relationships. We set the target FDR level to $q = 0.01$ and compare the inferred graph against the ground truth.

The inferred graphs in Figure 2 show that our method accurately recovers the true summary causal graph, effectively eliminating spurious causal edges introduced by the over-specified lag order. A single false positive is observed in lag 3, which is likely due to the model not fully capturing the temporal relationship in lag 2, leaving some dependencies to be attributed to lag 3. In addition, a small number of false negatives appear along the diagonal entries at lags 2 and 3, a behavior consistent with the DGP, as the corresponding coefficients are deliberately set to be small to reflect the decay of autocorrelations over longer lags. Overall, these results provide empirical support for the effectiveness of the proposed method.

### 4.4. Sensitivity Analysis

As discussed in Appendix A, component-wise modeling is often a structural compromise adopted to enable Granger causality identification via input layer weights. However, multivariate time series are inherently generated by complex dynamical systems with interdependencies. Consequently, modeling components separately may overlook shared dynamics, yielding suboptimal modeling performance. Leveraging the model-agnostic nature of our framework, we explicitly quantify the performance trade-off imposed by this modeling constraint. We conduct a sensitivity analysis on the first five subjects of the fMRI dataset, comparing *joint modeling* against *component-wise modeling* using both MLP and LSTM backbones. To ensure fairness, we control for model complexity by matching the number of trainable parameters across architectures. We benchmark our methods and Jacobian-based approach against the layer-weight method (Tank et al., 2021). The comparative results are summarized in Table 5.

There are two key observations from Table 5. First, across both MLP and LSTM backbones, joint modeling consistently outperforms its component-wise counterpart for all

regularizers. This performance disparity exposes the fundamental limitation of component-wise modeling: it fails to capture the shared dynamics and interdependencies inherent in multivariate time series, which are essential for accurate Granger causality identification. Second, within the component-wise regime, model-agnostic regularizers significantly outperform the layer-weight baseline. This result directly critiques the structural compromise discussed earlier: it demonstrates that relying on input layer weights as importance measure, the primary motivation for adopting component-wise models, yields inferior performance compared to model-agnostic approaches. Notably, our proposed Shap achieves the best overall performance. Collectively, these results validate the hypothesis that joint modeling enhances model expressiveness and demonstrate the versatility of our framework across different architectures.

### 5. Conclusion

In this work, we propose the Information-Theoretic Shapley value (Info-Shap), a novel measure for global feature importance. Theoretically, we establish that under the Info-Shap formulation, feature attribution is formally equivalent to Granger causality. Building upon this, we construct two novel regularization penalties, Shap and F-Shap, to promote structural sparsity and mitigate overfitting in NGC models. Extensive experiments on both synthetic and realistic datasets demonstrate the effectiveness of our method for temporal causal discovery, particularly in complex, high-dimensional scenarios.

Future research may extend this framework to accommodate latent confounders and contemporaneous effect by integrating Info-Shap into other model architectures. Furthermore, given the central role of importance measures highlighted in this work, exploring alternative importance metrics within this context is a promising direction. Lastly, the proposed regularizers can be adapted to other domains to improve model generalization and interpretability.

## Acknowledgements

The authors are grateful to the reviewers, Area Chairs, and Program Committee members for their constructive comments and suggestions, which have significantly improved the paper. This work is supported by the Zhongguancun Academy (nos. C20250504, XTS0006 and XTS0043). Zhoufan Zhu's work is supported by the National Natural Science Foundation of China (nos. 72503203 and 72573142), the Natural Science Foundation of Fujian Province (no. 2025J08009), and the Fundamental Research Funds for the Central Universities (no. 20720251053). Muyi Li's work is supported by the National Natural Science Foundation of China (nos. 72073112, 72033008, and 71988101), the Natural Science Foundation of Fujian Province (no. 2025J01036), the Fujian Key Laboratory of Statistics, and the Xiamen University Laboratory of Digital Finance.

## Impact Statement

This paper presents work whose goal is to advance the field of Machine Learning. There are many potential societal consequences of our work, none which we feel must be specifically highlighted here.

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

# A. Additional Details on Related Work

A fundamental insight underpinning this work is that a wide range of existing NGC methods can be synthesized through a unified framework based on feature attribution. From this perspective, many NGC approaches are, at their core, attempts to address the interpretability challenge inherent to neural networks. While classical linear models offer direct interpretability through the coefficients, neural networks operate as "black box" universal function approximators, and the weight matrices lack intrinsic interpretability (Zhang et al., 2021). Consequently, to establish a connection between neural networks and Granger causality, existing methodologies predominantly rely on imposing structural constraints on the network architecture. These constraints make specific components of the network to be "interpretable", thereby facilitating the evaluation of feature importance to infer causality.

We formalize this observation by examining prominent NGC methods through the lens of feature attribution. This unified perspective elucidates the mechanisms underlying current approaches and characterizes the limitations associated with relying on local importance measures.

**The Classical VAR Model.** We begin by revisiting the classical VAR model (Shojaie & Fox, 2022), which serves as the canonical example where the feature importance is explicitly parameterized. Let $\{\mathbf{X}_t \in \mathbb{R}^d\}_{t=1}^T$ be a $d$ dimensional multivariate time series. The VAR model of order $K$ is defined as:

$$\mathbf{X}_t = \sum_{k=1}^K A_k \mathbf{X}_{t-k} + \varepsilon_t, \tag{16}$$

where $A_k \in \mathbb{R}^{d \times d}$ is the coefficient matrix at lag $k$ and $\varepsilon_t$ is an independent error term. In this formulation, the $(i,j)$-th entry of the coefficient matrix $A_k$ quantifies the linear dependence of $X_t^i$ on $X_{t-k}^j$. Specifically, $A_{k,ij} = 0$ indicates that the $k$-th lag of series $j$ does not contribute to the prediction of series $i$, whereas a larger magnitude of $|A_{k,ij}|$ reflects a stronger linear dependence. By Definition 2.1, if $A_{k,ij} = 0$ for all $k = 1, \ldots, K$, then series $j$ does not Granger cause series $i$.

In the VAR framework, feature importance is characterized by the model parameters. Mathematically, these coefficients correspond to the first-order partial derivatives (Jacobian) of the output with respect to the input, fully capturing dependencies under the linear assumption. However, real-world systems often exhibit complex, nonlinear interactions that linear models cannot capture. To address this, many NGC methods generalize the VAR formulation by modeling such nonlinearities with neural networks.

**Constraint Joint Modeling.** A direct extension of the VAR framework involves parameterizing the coefficients as input-dependent functions to capture non-linear dependencies. For instance, Marcinkevičs & Vogt (2021) proposed using self-explaining neural networks (SENN) to estimate the time-varying coefficients. Their method is formulated as:

$$\mathbf{X}_t = \sum_{k=1}^p g_{\theta_k}(\mathbf{X}_{t-k}) \mathbf{X}_{t-k} + \varepsilon_t, \tag{17}$$

where $g_{\theta_k}(\cdot) : \mathbb{R}^d \to \mathbb{R}^{d \times d}$ is a neural network parameterized by $\theta_k$. Analogous to the VAR model, this approach adopt the coefficient matrices $g_{\theta_k}(\cdot)$ as the feature importance measure. However, we argue that examining $g_{\theta_k}(\cdot)$ provides *only a necessary, but not sufficient*, condition for identifying Granger causality in this non-linear setting.

To see this explicitly, consider the derivative of the $i$-th output with respect to the $j$-th input at lag $k$:

$$\frac{\partial X_t^i}{\partial X_{t-k}^j} = [g_{\theta_k}(\mathbf{X}_{t-k})]_{ij} + \sum_{m=1}^d \frac{\partial [g_{\theta_k}(\mathbf{X}_{t-k})]_{im}}{\partial X_{t-k}^j} X_{t-k}^m, \tag{18}$$

where we use $[g_{\theta_k}(\mathbf{X}_{t-k})]_{ij}$ to denote the $(i,j)$-th entry of the matrix $g_{\theta_k}(\mathbf{X}_{t-k})$. As derived above, the first term corresponds to the interpretable coefficient, while the second term captures the interaction between the input and the coefficient function. Consequently, enforcing sparsity on $g_{\theta_k}(\cdot)$ alone does not imply that the output is invariant with respect to the input. Thus, relying solely on the coefficient matrix yields an incomplete characterization of the underlying causal mechanism. This limitation is shared by many subsequent works (e.g., Bussmann et al. (2021); Han et al. (2025)), which adopt an analogous strategy of constraining specific network components to ensure interpretability.

**Component-wise Modeling.** An alternative strategy to achieve interpretability involves the use of component-wise architectures, which explicitly disentangle the influence of the input on each output dimension (Tank et al., 2021). Formally,

the model for the $i$-th dimension is defined as

$$\mathbf{X}_t^i = g^i(W_1\widetilde{\mathbf{X}}_{t-1}) + \varepsilon_{i,t}, \tag{19}$$

where $\widetilde{\mathbf{X}}_{t-1}$ is the lagged feature vector (Section 3.4), $W_1 \in \mathbb{R}^{h \times Kd}$ denotes the first layer weight matrix, and $g^i(\cdot) : \mathbb{R}^u \to \mathbb{R}$ represents the subsequent non-linear layers of the network with hidden dimension $h$. In this simple setting, Granger causality is identified by examining the sparsity pattern of $W_1$. Specifically, forcing the columns of $W_1$ associated with a given input series to zero prevents that series from influencing the output. Furthermore, importance measure $W_1$ in this framework is primarily topological rather than metric: it serves as a binary indicator for the existence of a causal link, rather than a direct measure of its magnitude. As noted by Tank et al. (2021), this criterion constitutes only a necessary condition for Granger non-causality.

This mechanism is structurally analogous to the VAR model, in which zero coefficients exclude specific lagged dependencies. Same as constrained joint modeling, rely on the interpretable part is inadequate, as the output-input derivative also involves the non-linear transformation $g^i(\cdot)$. Owing to its conceptual simplicity, this paradigm has been widely adopted in subsequent work, including Khanna & Tan (2020), Suryadi et al. (2023), and Poonia et al. (2025).

**Discussion.** As demonstrated, existing NGC methods typically necessitates imposing structural constraints on the model architecture to achieve interpretability, thereby bridge the gap between the black-box modeling and causal discovery. These methods derive interpretability from specific model components, such as coefficients in linear-like formulations or weight matrices in restricted network modules. However, we argue that relying solely on these interpretable components yields an insufficient characterization of the complex functional relationships between inputs and outputs. While exceptions like JRNGC (Zhou et al., 2024) circumvent architectural constraints by employing the input-output Jacobian, they remain limited by the local nature of first-order approximations, which fail to capture the interactions among features.

The fundamental insight of this work is grounded in the structural triad of our feature attribution framework (Figure 1). We observe that while significant attention has been dedicated to model architecture design, the role of the importance measure remains under-explored, leading to a misalignment between model expressiveness and structural inference mechanism. Consequently, our method seeks interpretability external to the model parameters, rather than forcing the model to be intrinsically transparent. We hope this work serves to re-align the focus of the literature, drawing greater attention to the development of rigorous importance measures as a critical complement to advanced model architectures.

# B. Derivations and Proofs

## B.1. Detailed Derivations

**Derivation of CGIM under Gaussian Assumption.** We begin by recalling the definition of the Conditional Generative Information Matrix (CGIM):

$$\mathrm{CGIM}_{Y|\mathbf{X}} = \mathbb{E}_{Y|\mathbf{X}} \left[ \nabla_{\mathbf{X}} \log p(Y|\mathbf{X}) \nabla_{\mathbf{X}} \log p(Y|\mathbf{X})^\top \right], \tag{20}$$

where $Y \in \mathbb{R}$ and $\mathbf{X} \in \mathbb{R}^d$ denote the scalar output and vector input, respectively. Under the Gaussian assumption $Y|\mathbf{X} \sim \mathcal{N}(g(\mathbf{X}), \sigma^2)$, the conditional probability density function is given by:

$$p(Y|\mathbf{X}) = \frac{1}{\sqrt{2\pi\sigma^2}} \exp\left( -\frac{1}{2\sigma^2}(Y - g(\mathbf{X}))^2 \right). \tag{21}$$

To derive the score function, we first write the log-likelihood:

$$\log p(Y|\mathbf{X}) = -\frac{1}{2}\log(2\pi\sigma^2) - \frac{1}{2\sigma^2}(Y - g(\mathbf{X}))^2. \tag{22}$$

Taking the gradient with respect to the input $\mathbf{X}$ yields the score function:

$$\nabla_{\mathbf{X}} \log p(Y|\mathbf{X}) = \frac{Y - g(\mathbf{X})}{\sigma^2}\partial g(\mathbf{X}), \tag{23}$$

where $\partial g(\mathbf{X})$ denotes the gradient of the mean function $g$ with respect to $\mathbf{X}$. Substituting this expression into the definition of CGIM in (20), we obtain:

$$
\begin{aligned}
\mathrm{CGIM}_{Y|\mathbf{X}} &= \mathbb{E}_{Y|\mathbf{X}} \left[ \left( \frac{Y - g(\mathbf{X})}{\sigma^2} \partial g(\mathbf{X}) \right) \left( \frac{Y - g(\mathbf{X})}{\sigma^2} \partial g(\mathbf{X}) \right)^\top \right] \\
&= \frac{1}{\sigma^4} \partial g(\mathbf{X}) \underbrace{\mathbb{E}_{Y|\mathbf{X}} \left[ (Y - g(\mathbf{X}))^2 \right]}_{\mathrm{Var}(Y|\mathbf{X}) = \sigma^2} \partial g(\mathbf{X})^\top \\
&= \frac{1}{\sigma^2} \partial g(\mathbf{X}) \partial g(\mathbf{X})^\top.
\end{aligned}
\tag{24}
$$

**Derivation of Equation** (9). We begin with the formula of Info-Shap (5):

$$
\phi_i(v_{\mathrm{Info}}) = \mathbb{E} \left[ |\partial_i g|^2 \right] + \frac{1}{2} \sum_{j \neq i} \mathbb{E} \left[ |\partial_i g \, \partial_j g| \right].
\tag{25}
$$

The Shapley regularizer is defined as follows:

$$
\begin{aligned}
\Omega_{\mathrm{Shap}}(g; \lambda) &= \lambda \sum_{i=1}^{d} |\phi_i(v_{\mathrm{Info}})| \\
&= \lambda \sum_{i=1}^{d} \left( \mathbb{E}[|\partial_i g|^2] + \frac{1}{2} \sum_{j \neq i} \mathbb{E}[|\partial_i g \, \partial_j g|] \right).
\end{aligned}
\tag{26}
$$

By vectorizing the inner sums, we have:

$$
\begin{aligned}
\Omega_{\mathrm{Shap}}(g; \lambda) &= \lambda \sum_{i=1}^{d} \left( \frac{1}{2} \mathbb{E}[|\partial_i g|^2] + \frac{1}{2} \sum_{j=1}^{d} \mathbb{E}[|\partial_i g \, \partial_j g|] \right) \\
&= \frac{\lambda}{2} \mathbb{E} \left( \sum_{i=1}^{d} |\partial_i g|^2 + \sum_{i=1}^{d} \sum_{j=1}^{d} |\partial_i g \, \partial_j g| \right) \\
&= \frac{\lambda}{2} \left( \mathbb{E} \|\partial g\|_2^2 + \mathbb{E} \|\partial g \partial g^\top\|_{\ell_1} \right).
\end{aligned}
\tag{27}
$$

### B.2. Multivariate Extension of Info-Shap and Regularizers

This section extends the Info-Shap framework to the multivariate output setting, where $\mathbf{Y} \in \mathbb{R}^{d_{out}}$. Let $g : \mathbb{R}^{d_{in}} \to \mathbb{R}^{d_{out}}$ denote a differentiable function with Jacobian matrix $\mathbf{J}(\mathbf{X}) \in \mathbb{R}^{d_{out} \times d_{in}}$. The entry $\mathbf{J}_{ij}(\mathbf{X}) = \partial g_i(\mathbf{X})/\partial X^j$ represents the partial derivative of the $i$-th output component with respect to the $j$-th input feature. Accordingly, the Info-Shap value for feature $j$ relative to output $i$ is defined as:

$$
\phi_{ij}(v_{\mathrm{Info}}) = \mathbb{E}_{\mathbf{X}} \left[ \mathbf{J}_{ij}^2(\mathbf{X}) \right] + \frac{1}{2} \sum_{k \neq j} \mathbb{E}_{\mathbf{X}} \left[ |\mathbf{J}_{ij}(\mathbf{X}) \mathbf{J}_{ik}(\mathbf{X})| \right].
\tag{28}
$$

The multivariate Shapley regularizers (Shap) aggregates these importance scores across all output dimensions. By vectorizing the outer product of the row-wise gradients, we formulate Shap as:

$$
\begin{aligned}
\Omega_{\mathrm{Shap}}(g; \lambda) &= \lambda \sum_{i=1}^{d_{out}} \sum_{j=1}^{d_{in}} \phi_{ij}(v_{\mathrm{Info}}) \\
&= \frac{\lambda}{2} \sum_{i=1}^{d_{out}} \mathbb{E}_{\mathbf{X}} \left[ \|\mathbf{J}_{i,:}(\mathbf{X})\|_2^2 + \|\mathbf{J}_{i,:}^\top(\mathbf{X}) \mathbf{J}_{i,:}(\mathbf{X})\|_{\ell_1} \right], \\
&= \frac{\lambda}{2} \mathbb{E}_{\mathbf{X}} \left[ \|\mathbf{J}_{i,:}(\mathbf{X})\|_F^2 + \sum_{i=1}^{d_{out}} \|\mathbf{J}_{i,:}^\top(\mathbf{X}) \mathbf{J}_{i,:}(\mathbf{X})\|_{\ell_1} \right]
\end{aligned}
\tag{29}
$$

where $\mathbf{J}_{i,:}$ denotes the $i$-th row of the Jacobian.

For the F-Shap variant, we replace the element-wise $\ell_1$ norm to squared Frobenius norm, which facilitates a computationally efficient vectorized formulation:

$$\Omega_{\text{F-Shap}}(g; \lambda_1, \lambda_2) = \frac{1}{2}\mathbb{E}_{\mathbf{X}}\left[\lambda_1\|\mathbf{J}_{i,:}(\mathbf{X})\|_F^2 + \lambda_2 \sum_{i=1}^{d_{out}} \|\mathbf{J}_{i,:}^\top(\mathbf{X})\mathbf{J}_{i,:}(\mathbf{X})\|_F^2\right]. \tag{30}$$

### B.3. Proof of Theorem 3.1

We begin by defining the Möbius transform of set functions, a necessary prerequisite for invoking established combinatorial results from the literature.

**Definition B.1** (Möbius Transform). Let $N$ be a finite set of players. Given a set function $v : 2^N \to \mathbb{R}$, its Möbius transform $m : 2^N \to \mathbb{R}$ is defined as:

$$m(S) = \sum_{T \subseteq S} (-1)^{|S|-|T|}v(T), \quad \forall S \subseteq N. \tag{31}$$

The original set function can be uniquely recovered via the inverse Möbius transform:

$$v(S) = \sum_{T \subseteq S} m(T), \quad \forall S \subseteq N. \tag{32}$$

**Remark.** The Möbius transform provides an alternative representation of set functions, analogous to the Fourier transform for continuous functions. It decomposes $v(S)$ into **orthogonal incremental components**, where each coefficient $m(T)$ represents the net contribution of coalition $T$ that cannot be explained by any of its proper sub-coalitions.

We leverage the following theorem from Grabisch (1997) to establish the connection between the Shapley value and the Möbius coefficients:

**Proposition B.2** (Grabisch, 1997; Theorem 1). *The Shapley value $\phi_i(v)$ can be expressed as a weighted sum of its Möbius coefficients:*

$$\phi_i(v) = \sum_{S \subseteq N : i \in S} \frac{m(S)}{|S|}, \quad \forall i \in N. \tag{33}$$

*Proof.* The proof proceeds by deriving the Möbius coefficients for the Info-Shap set function $v_{\text{Info}}$ and then substituting them into (33).

Recall the definition of the set function:

$$v_{\text{Info}}(S) = \sum_{i \in S} \mu_{ii} + \sum_{\{i,j\} \subseteq S, i<j} \mu_{ij}, \tag{34}$$

where $\mu_{ij} = \mathbb{E}[|\partial_i g\, \partial_j g|]$. Since $v_{\text{Info}}$ involves interactions of at most order two, it represents a **2-additive capacity** (Grabisch, 1997). We therefore derive the Möbius coefficients $m(S)$ by cardinality of $S$:

**Case 1: First-order** ($|S| = 1$). Let $S = \{i\}$.

$$m(i) = (-1)^{1-1}v(i) = \mu_{ii} = \mathbb{E}[(\partial_i g)^2]. \tag{35}$$

**Case 2: Second-order** ($|S| = 2$). Let $S = \{i, j\}$ with $i \neq j$.

$$\begin{aligned} m(i,j) &= v(i,j) - v(i) - v(j) + v(\emptyset) \\ &= (\mu_{ii} + \mu_{jj} + \mu_{ij}) - \mu_{ii} - \mu_{jj} + 0 \\ &= \mu_{ij} = \mathbb{E}[|\partial_i g \partial_j g|]. \end{aligned} \tag{36}$$

**Case 3: Higher-order** ($|S| > 2$). Since $v_{\text{Info}}$ is 2-additive, all higher-order Möbius coefficients vanish:

$$m(S) = 0, \quad \forall |S| > 2. \tag{37}$$

Thus, the Möbius coefficients for $v_{\text{Info}}$ are fully characterized by the pairwise gradient expectations:

$$m(S) = \begin{cases} \mathbb{E}[(\partial_i g)^2] & \text{if } S = i, \\ \mathbb{E}[|\partial_i g \partial_j g|] & \text{if } S = i, j, i \neq j, \\ 0 & \text{otherwise.} \end{cases} \tag{38}$$

Substituting (38) into (33), the summation reduces to terms where $|S| = 1$ and $|S| = 2$:

$$\begin{aligned} \phi_i(v_{\text{Info}}) &= \sum_{S \subseteq N : i \in S} \frac{m(S)}{|S|} \\ &= \mathbb{E}[(\partial_i g)^2] + \frac{1}{2} \sum_{j \neq i} \mathbb{E}[|\partial_i g \partial_j g|]. \end{aligned} \tag{39}$$

This concludes the proof. □

### B.4. Proof of Theorem 3.2

*Proof.* Recall the Info-Shap for feature $X^i$ is given by:

$$\phi_i(v_{\text{Info}}) = \mathbb{E}[(\partial_i g)^2] + \frac{1}{2} \sum_{j \neq i} \mathbb{E}[|\partial_i g \, \partial_j g|]. \tag{40}$$

We establish the equivalence by proving the implications in both directions.

**Sufficiency.** ($\impliedby$) Assume the $i$-th feature $X^i$ is conditionally mean independent of $Y$ given the remaining features $\mathbf{X}^{-i}$. By definition, this implies that the conditional mean function satisfies:

$$\mathbb{E}[Y \mid \mathbf{X}] = \mathbb{E}[Y \mid \mathbf{X}^{-i}]. \tag{41}$$

Since the regression function is defined as $g(\mathbf{X}) = \mathbb{E}[Y \mid \mathbf{X}]$, $g(\cdot)$ depends solely on $\mathbf{X}^{-i}$ and is constant with respect to feature $X^i$. Consequently, the partial derivative with respect to $X^i$ equals zero almost surely:

$$\partial_i g(\mathbf{X}) = 0, \quad \text{a.s.} \tag{42}$$

Substituting this into (40), every term in the summation contains the factor $\partial_i g$. Thus:

$$\phi_i(v_{\text{Info}}) = \mathbb{E}[0] + \frac{1}{2} \sum_{j \neq i} \mathbb{E}[0 \cdot |\partial_j g|] = 0. \tag{43}$$

**Necessity.** ($\implies$) Assume that the Info-Shap value is zero, i.e., $\phi_i(v_{\text{Info}}) = 0$. We observe that (40) is a sum of non-negative terms. For the total sum to be zero, every individual term must be identically zero. In particular, the first term must satisfy:

$$\mathbb{E}[(\partial_i g)^2] = 0. \tag{44}$$

Since $(\partial_i g)^2 \geq 0$, a zero expectation implies that the random variable itself is zero almost surely:

$$\partial_i g(\mathbf{X}) = 0, \quad \text{a.s.} \tag{45}$$

A zero gradient implies that the function $g(\mathbf{X})$ is constant with respect to the coordinate $X^i$. Therefore, the conditional expectation $g(\mathbf{X}) = \mathbb{E}[Y \mid \mathbf{X}]$ is functionally invariant to $X^i$. This recovers the definition of conditional mean independence:

$$\mathbb{E}[Y \mid \mathbf{X}] = \mathbb{E}[Y \mid \mathbf{X}^{-i}]. \tag{46}$$

The proof is complete. □

## C. Algorithmic Details

### C.1. Stochastic Approximation of Fast-Shapley Regularizer

We leverage Hutchinson's trace estimator (Hutchinson, 1989) to efficiently approximate the regularizer, thereby avoiding the explicit instantiation of the full Jacobian matrix $\mathbf{J}(\mathbf{x})$.

**Hutchinson's estimator.** For any square matrix $\mathbf{M}$, its trace can be unbiasedly estimated as $\mathrm{tr}(\mathbf{M}) = \mathbb{E}[\mathbf{z}^\top \mathbf{M}\mathbf{z}]$, where $\mathbf{z}$ is a random projection vector with zero mean and identity covariance (e.g., drawn from a Rademacher or Gaussian distribution). In practice, we approximate this expectation using $R$ random projections:

$$\mathrm{tr}(\mathbf{M}) \approx \frac{1}{R} \sum_{r=1}^{R} \mathbf{z}_r^\top \mathbf{M}\mathbf{z}_r. \tag{47}$$

We apply this method to both components of $\Omega_{\text{F-Shap}}$ (30) by reformulating them into trace operations.

**1. Individual Term (via JVP).** The individual effect corresponds to the squared Frobenius norm $\|\mathbf{J}(\mathbf{x})\|_F^2$. To align with the interaction term estimation (see below) and maximize computational efficiency, we express this as $\mathrm{tr}(\mathbf{J}^\top \mathbf{J})$. By sampling random vectors $\mathbf{z}_r \in \mathbb{R}^{d_{in}}$ from the input space, we obtain the estimator:

$$\|\mathbf{J}(\mathbf{x})\|_F^2 = \mathrm{tr}(\mathbf{J}^\top \mathbf{J}) \approx \frac{1}{R} \sum_{r=1}^{R} \mathbf{z}_r^\top (\mathbf{J}^\top \mathbf{J})\mathbf{z}_r = \frac{1}{R} \sum_{r=1}^{R} \|\mathbf{J}(\mathbf{x})\mathbf{z}_r\|_2^2. \tag{48}$$

We define the intermediate vector $\mathbf{u}_r \in \mathbb{R}^{d_{out}}$ as the **Jacobian-Vector Product (JVP)**:

$$\mathbf{u}_r(\mathbf{x}) = \mathbf{J}(\mathbf{x})\mathbf{z}_r = \nabla_\epsilon g(\mathbf{x} + \epsilon \mathbf{z}_r)\big|_{\epsilon=0}. \tag{49}$$

In automatic differentiation frameworks (e.g., PyTorch, JAX), $\mathbf{u}_r(\mathbf{x})$ corresponds to a standard forward-mode differentiation pass or can be efficiently computed via the double-backward trick.

**2. Interaction Term (via Double JVP).** The interaction term corresponds to the sum of the squared Frobenius norms of the rank-1 interaction matrices for each output dimension. Let $\mathbf{A}_i = \mathbf{J}_{i,:}^\top \mathbf{J}_{i,:}$ denote the outer product of the gradient for the $i$-th output. The regularization term is given by:

$$\sum_{i=1}^{d_{out}} \|\mathbf{A}_i\|_F^2 = \sum_{i=1}^{d_{out}} \mathrm{tr}\left(\mathbf{A}_i \mathbf{A}_i^\top\right). \tag{50}$$

Since a single random projection cannot unbiasedly estimate the squared trace of a rank-1 matrix (Avron & Toledo, 2011), we employ a "double-sample" estimator. Let $\mathbf{z}_{r,1}, \mathbf{z}_{r,2} \in \mathbb{R}^{d_{in}}$ be two independent random vectors. We approximate the trace as:

$$\begin{aligned}
\mathrm{tr}(\mathbf{A}_i \mathbf{A}_i^\top) &\approx \frac{1}{R} \sum_{r=1}^{R} \left(\mathbf{z}_{r,1}^\top \mathbf{A}_i \mathbf{z}_{r,2}\right)^2 \\
&= \frac{1}{R} \sum_{r=1}^{R} \left[(\mathbf{z}_{r,1}^\top \mathbf{J}_{i,:}^\top)(\mathbf{J}_{i,:}\mathbf{z}_{r,2})\right]^2.
\end{aligned} \tag{51}$$

Using the JVP notation $\mathbf{u}_{r,1} = \mathbf{J}(\mathbf{x})\mathbf{z}_{r,1}$ and $\mathbf{u}_{r,2} = \mathbf{J}(\mathbf{x})\mathbf{z}_{r,2}$, the estimator simplifies to the dot product of the element-wise squared vectors:

$$\sum_{i=1}^{d_{out}} \|\mathbf{A}_i\|_F^2 \approx \frac{1}{R} \sum_{r=1}^{R} \sum_{i=1}^{d_{out}} ([\mathbf{u}_{r,1}]_i [\mathbf{u}_{r,2}]_i)^2 = \frac{1}{R} \sum_{r=1}^{R} \langle \mathbf{u}_{r,1}^2, \mathbf{u}_{r,2}^2 \rangle, \tag{52}$$

where the square operation on vectors is applied element-wise.

**Total Regularizer Approximation.** We approximate $\Omega_{\text{F-Shap}}(g; \lambda_1, \lambda_2)$ by averaging over the dataset (or mini-batch) and $R$ random projection pairs. For each pair $r$, we compute $\mathbf{u}_{r,1}$ and $\mathbf{u}_{r,2}$ and combine them. Note that we reuse these JVPs to estimate the individual term:

$$\widehat{\Omega}_{\text{F-Shap}}(g) = \frac{1}{2T'R} \sum_{t} \sum_{r=1}^{R} \left[\lambda_1 \frac{\|\mathbf{u}_{r,1}\|_2^2 + \|\mathbf{u}_{r,2}\|_2^2}{2} + \lambda_2 \langle \mathbf{u}_{r,1}^2, \mathbf{u}_{r,2}^2 \rangle\right]. \tag{53}$$

**Computational Complexity Analysis.** To demonstrate the scalability of our approach, we compare the computational complexity of the approximated computation against the exact one. Let $\mathcal{C}_g$ denote the cost of a single backward pass (or equivalently, a Jacobian-Vector Product). The exact computation of $\Omega_{\text{F-Shap}}$ entails instantiating the full Jacobian $\mathbf{J} \in \mathbb{R}^{d_{out} \times d_{in}}$, which requires $d_{out}$ distinct backward passes—one for each output dimension. Consequently, the total complexity scales linearly as $\mathcal{O}(d_{out} \cdot \mathcal{C}_g)$. In settings with high-dimensional outputs, this overhead becomes prohibitive for iterative training. In contrast, the random projection approximation decouples the computational cost from the output dimension. By requiring only $2R$ JVP operations per iteration, it achieves a complexity of $\mathcal{O}(R \cdot \mathcal{C}_g)$. Regarding the number of random projections $R$, Hoffman et al. (2019) demonstrated that for Jacobian regularization with standard mini-batch sizes (e.g., $B = 100$), a small number of projections (even $R = 1$) often yields performance comparable to the exact full Jacobian. We provide empirical validation of these efficiency gains and a sensitivity analysis of $R$ in Section D.3.

---

**Algorithm 1** Approximated Fast-Shapley Regularization

---

1: **Input:** Minibatch $\mathbf{X} \in \mathbb{R}^{B \times d_{in}}$, Model $g(\cdot)$, Projections $R$, Hyperparams $\lambda_1, \lambda_2$.
2: **Output:** Estimate $\widehat{\Omega}_{\text{F-Shap}}$.
3: Initialize penalty accumulator $\mathcal{L}_{\text{F-Shap}} \leftarrow 0$.
4: **for** $r = 1$ **to** $R$ **do**
5:     *// Sample noise for the entire batch (Shape: $B \times d_{in}$)*
6:     Sample $\mathbf{Z}_1, \mathbf{Z}_2$ independently from Rademacher distribution.
7:     *// Compute JVP for entire batch (Shape: $B \times d_{out}$)*
8:     $\mathbf{U}_1 = \text{JVP}(g, \mathbf{X}, \mathbf{Z}_1)$
9:     $\mathbf{U}_2 = \text{JVP}(g, \mathbf{X}, \mathbf{Z}_2)$
10:     *// 1. Individual Term: Row-wise squared $\ell_2$ norms*
11:     $\mathbf{n}_1 = \text{sum}(\mathbf{U}_1 \odot \mathbf{U}_1, \text{dim} = 1)$
12:     $\mathbf{n}_2 = \text{sum}(\mathbf{U}_2 \odot \mathbf{U}_2, \text{dim} = 1)$
13:     $L_{\text{ind}} = 0.5 \cdot (\text{mean}(\mathbf{n}_1) + \text{mean}(\mathbf{n}_2))$
14:     *// 2. Interaction Term: Row-wise dot product of squared matrices*
15:     $\mathbf{S}_1 = \mathbf{U}_1 \odot \mathbf{U}_1$    *// Element-wise square*
16:     $\mathbf{S}_2 = \mathbf{U}_2 \odot \mathbf{U}_2$
17:     $L_{\text{inter}} = \text{mean}(\text{sum}(\mathbf{S}_1 \odot \mathbf{S}_2, \text{dim} = 1))$
18:     $\mathcal{L}_{\text{F-Shap}} \leftarrow \mathcal{L}_{\text{F-Shap}} + \lambda_1 L_{\text{ind}} + \lambda_2 L_{\text{inter}}$
19: **end for**
20: **return** $\mathcal{L}_{\text{F-Shap}}/R$.

---

### C.2. Shapley Regularized Neural Granger Causality

We present the complete training procedure for the proposed SRNGC and F-SRNGC methods in Algorithm 2.

## D. Additional Experiments

### D.1. Dataset Description

**VAR(3) Model.** We begin with the Vector Autoregressive (VAR) model, a standard linear benchmark for Granger causality identification (Shojaie & Fox, 2022). We set the lag order to $K = 3$, with data generated according to the following equation:

$$\mathbf{X}_t = A_1 \mathbf{X}_{t-1} + A_2 \mathbf{X}_{t-2} + A_3 \mathbf{X}_{t-3} + \varepsilon_t, \tag{54}$$

where $\mathbf{X}_t \in \mathbb{R}^d$ denotes the vector at time $t$, $A_k \in \mathbb{R}^{d \times d}$ are the coefficient matrices, and $\varepsilon_t \in \mathbb{R}^d \sim \mathcal{N}(0, \mathbf{I}_d)$ represents i.i.d. Gaussian noise. The coefficient matrices $A_1, A_2, A_3$ are randomly generated such that the overall sparsity of the ground-truth causal graph is 0.3. For each configuration of dimension $d \in \{10, 50\}$ and length $T \in \{500, 1000\}$, we generate five independent realizations.

**Lorenz-96 System.** To evaluate performance on nonlinear dynamics, we employ the Lorenz-96 system, a canonical model for atmospheric circulation known for its chaotic behavior (Karimi & Paul, 2010). The dynamics are governed by the set of differential equations:

$$\frac{dX_t^i}{dt} = (X_t^{i+1} - X_t^{i-2})X_t^{i-1} - X_t^i + F, \tag{55}$$

---

**Algorithm 2** Shapley Regularized Neural Granger Causality (SRNGC)

1: **Input:** Multivariate time series $\{\mathbf{X}_t\}_{t=1}^{T} \in \mathbb{R}^d$, Lag order $K$, Regularization strength $\lambda$, FDR level $q$, Learning rate $\eta$.
2: **Output:** Estimated Granger Causal Graph $\mathcal{G}$ (Adjacency Matrix $\mathbf{A} \in \{0,1\}^{d \times d}$).
3: *// Phase 1: Model Fitting & Residual Calculation*
4: Construct lagged feature matrix $\widetilde{\mathbf{X}} \in \mathbb{R}^{T' \times Kd}$ and target $\mathbf{Y} \in \mathbb{R}^{T' \times d}$.
5: Initialize model $g_\theta(\cdot)$.
6: **while** not converged **do**
7:     Sample minibatch $(\widetilde{\mathbf{x}}, \mathbf{y})$.
8:     Compute Loss: $\mathcal{L}(\theta) = \|\mathbf{y} - g_\theta(\widetilde{\mathbf{x}})\|_2^2 + \widehat{\Omega}(g_\theta; \lambda)$.
9:     $\{$Use $\widehat{\Omega}_{\text{Shap}}$ or approximation $\widehat{\Omega}_{\text{F-Shap}}$ (Alg. 1).$\}$
10:     Update $\theta \leftarrow \theta - \eta \nabla_\theta \mathcal{L}(\theta)$.
11: **end while**
12: Compute residuals: $\hat{\mathcal{E}}_t = \mathbf{X}_t - g_\theta(\widetilde{\mathbf{X}}_{t-1})$ for $t = K+1 \ldots T$.
13: *// Phase 2: Model-X Knockoff Generation*
14: Generate permutation $\pi$ of indices $\{K+1, \ldots, T\}$.
15: Construct knockoff features: $\widetilde{\mathbf{X}}_{t-1}^{\text{ko}} = g_\theta(\widetilde{\mathbf{X}}_{t-1}) + \hat{\mathcal{E}}_{\pi(t)}$.
16: Construct augmented dataset $\mathcal{D}_{\text{aug}}$ with features $[\widetilde{\mathbf{X}}_{t-1}^\top, \widetilde{\mathbf{X}}_{t-1}^{\text{ko}\top}]^\top$.
17: *// Phase 3: Causal Discovery & Thresholding*
18: Retrain model $g'_\phi$ on $\mathcal{D}_{\text{aug}}$ using the same objective $\mathcal{L}$.
19: Compute Info-Shap importance scores using Eq. (13):
20:     $\widehat{\phi}_{ijk}^* \leftarrow \text{InfoShap}(g'_\phi, \text{Original Features})$
21:     $\widehat{\phi}_{ijk}^{\text{ko}} \leftarrow \text{InfoShap}(g'_\phi, \text{Knockoff Features})$
22: Calculate statistics: $W_{ijk} = \widehat{\phi}_{ijk}^* - \widehat{\phi}_{ijk}^{\text{ko}}$.
23: Determine FDR threshold $\alpha_q$:
24:     $\alpha_q = \min\{\alpha > 0 : \frac{1 + |\{(i,j,k):W_{ijk} \leq -\alpha\}|}{|\{(i,j,k):W_{ijk} > \alpha\}| \vee 1} \leq q\}$.
25: Construct Graph $\mathcal{G}$: $\mathbf{A}_{ij} = 1$ if $\exists k, W_{ijk} \geq \alpha_q$.
26: **return** Adjacency Matrix $\mathbf{A}$.

---

where $X_t^i$ represents the $i$-th component of the state vector and indices subject to cyclic boundary conditions (i.e., $X_t^{-1} = X_t^{d-1}$). We integrate the system using the fourth-order Runge-Kutta (RK4) method with a time step of $\Delta t = 0.01$. To simulate measurement noise, we add independent standard Gaussian noise ($\mathcal{N}(0,1)$) to the final trajectories. We fix the system dimension at $d = 50$ and examine two forcing regimes: $F = 10$ (weakly chaotic) and $F = 40$ (strongly chaotic). For each regime, we generate five independent realizations for every sequence length $T \in \{500, 1000\}$.

**DREAM3 Dataset.** We utilize data from the DREAM3 *In Silico* Network Challenge (Prill et al., 2010), a widely recognized benchmark for gene regulatory network reconstruction. This dataset consists of simulated gene expression time series derived from thermodynamic models of gene regulation. We focus on the *E. coli* (2 sequences) and *Yeast* (3 sequences) subnetworks, which pose significant challenges due to biological complexity and high noise levels. Crucially, this dataset represents a high-dimensional, limited-sample regime: for each sequence, it provides 46 independent realizations with dimension $d = 100$ and sequence length $T = 21$.

**CausalTime Dataset.** We further evaluate our method on CausalTime (Cheng et al., 2024b), a benchmark designed to generate time series that closely mimic real-world data distributions while providing ground-truth causal graphs. We adopt the published benchmark suite[1], comprising three datasets synthesized to emulate diverse empirical contexts: *Traffic* ($d = 20$), *Medical* ($d = 20$), and *AQI* ($d = 36$). Each dataset comprises 480 independent realizations, each with a sample length of $T = 40$.

**fMRI Dataset.** Finally, in sensitivity analysis, we apply our method to functional MRI (fMRI) data (Smith et al., 2011). This benchmark was designed to validate connectivity estimation methods against complex confounds inherent in BOLD signals, such as hemodynamic delays and inaccurate ROI definitions. We utilize the third dataset from this suite ($d = 15$), which contains the BOLD signals of 50 subjects, each with a sample length of $T = 200$. To evaluate robustness, we conduct

---

[1] https://www.causaltime.cc/

the comparison on the first five subjects and report the average performance along with standard deviations.

### D.2. Implementation Details

**Training Protocol.** We implemented the proposed framework using PyTorch. Prior to training, we standardized all time series to zero mean and unit variance, partitioning the data into training ($70\%$) and validation ($30\%$) sets. The models were optimized using the Adam algorithm to minimize Mean Squared Error (MSE). Training was conducted for a maximum of 2000 epochs, adopting an early stopping strategy with a patience of 50 epochs based on validation loss to prevent overfitting. For the stochastic approximation of the regularizers, we utilized random vectors drawn from a Rademacher distribution. After training, the empirical Info-Shap values (13) were computed using the full dataset to reduce estimation variance. All experiments were executed on a server running Ubuntu 20.04, equipped with an NVIDIA A800 GPU, 12 CPU cores, and 108 GB of RAM.

**Hyperparameter Configuration.** The hyperparameters of our methods contains the learning rate, regularization parameters $\lambda$ (or $\lambda_1$ and $\lambda_2$ for F-SRNGC), network parameters (e.g., number of hidden layers and hidden units) and dropout rate. We adopt the configuration protocol from Zhou et al. (2024), and further scaling the number of hidden units based on the dimension of data for realistic dataset. Full-batch training is employed for all experiments, with the exception of the CausalTime dataset, where the batch size is set to 512. Regarding the time lag $K$ for constructing lagged feature vector, we use the ground-truth for synthetic datasets and tune it via validation loss for realistic datasets. Final hyperparameters are selected based on the prediction loss on the validation set.

**Sensitivity Analysis Settings.** We adhere to the general training protocol described above, with specific adjustments to evaluate the method's sensitivity to model architecture and random projection settings. First, for the sensitivity analysis of random projections, we fix the model architecture to Residual MLP, explicitly setting the dropout rate to $0.0$ and the hidden dimension at $100$. Second, regarding the comparison between joint and component-wise modeling, we adjust the hidden layer dimensions to maintain comparable trainable parameter counts across architectures. Specifically, following Khanna & Tan (2020), we set the number of hidden units to $10$ for component-wise models and $100$ for joint models.

### D.3. Additional Sensitivity Analysis

**Computational Complexity.** We evaluate the computational overhead of random projection approximations relative to exact computation for both Jacobian and Shapley-based regularizers. Specifically, we benchmark the exact formulations (Jacob-F and F-Shap) against their stochastic approximations, denoted as Jacob-F($n$) and F-Shap($n$), where $n$ indicates the number of random projections used. Experiments are conducted on a fully connected MLP trained with mean squared error loss. To systematically assess the scaling behavior, we isolate the impact of model complexity (total trainable parameters) and data complexity (input dimensionality).

The experimental protocol isolates one configuration at a time while maintaining default values for the others: a hidden dimension of $128$, a depth of $2$ hidden layers, and a data dimension of $64$. To assess model complexity, we sweep the network depth from 1 to 20 layers and plot the runtime against the total number of trainable parameters. To assess data complexity, we vary the input dimension logarithmically from 10 to 1000, presenting the results on a log-log scale. For each configuration, we record the average per-epoch runtime over 100 independent epochs following a 20 epoch warm-up phase.

Figure 3 corroborates our complexity analysis. First, the exact implementations of both regularizers exhibit consistently high runtimes, driven by the computational burden of materializing the full Jacobian matrix. Consequently, these methods are highly sensitive to both model size (which governs $\mathcal{C}_g$) and data dimension (governing $d_{out}$). Second, regarding stochastic approximations, F-Shap incurs only marginal overhead relative to Jacob-F when using the same number of random projections. While the additional projection required to capture interaction effects introduces a small latency, this cost is negligible compared to full Jacobian instantiation. Regarding hyperparameter sensitivity, increasing the number of random projections scales the runtime and heightens sensitivity to model complexity. Nevertheless, all approximation versions maintain robustly constant runtimes as the data dimension increases. These results demonstrate that the approximation of F-Shap offers a scalable solution for high-dimensional settings where exact methods become prohibitive.

**Sensitivity to Random Projections.** We further investigate how the number of random projections $R$ affects the causal discovery performance of F-Shap. Using the first five subjects from the fMRI dataset, we evaluate configurations with $R \in \{1, 3, 5\}$, and denote the corresponding methods as F-Shap($R$). These variants are benchmarked against the exact Shap and a baseline without regularization. With Info-Shap as the importance measure, performance is assessed across

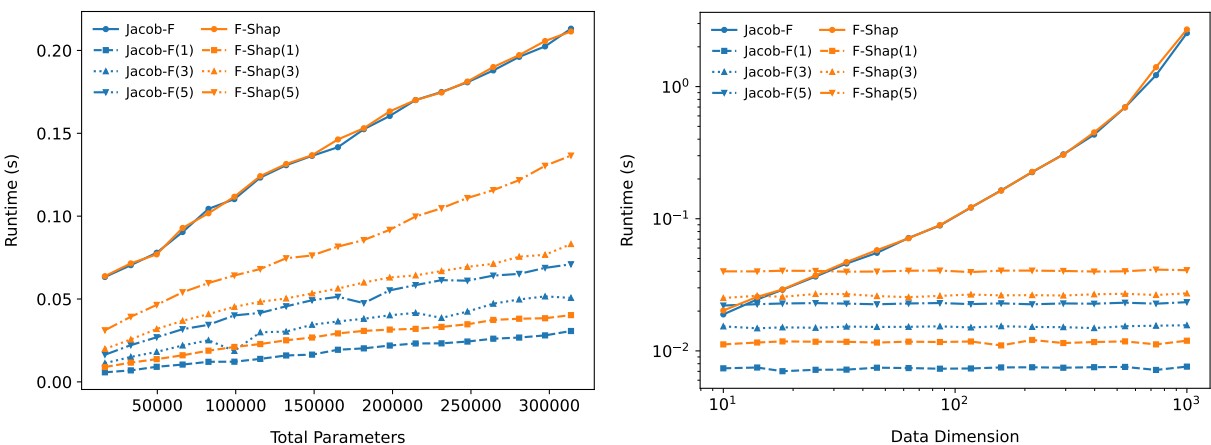

*Figure 3.* **Computational efficiency of causal regularizers. Left:** Runtime vs. model complexity (total trainable parameters), varying network depth from 1 to 20 layers (data dimension fixed at 64). **Right:** Runtime vs. data complexity (input dimension), plotted on a log-log scale (depth fixed at 2). In both cases, the hidden dimension is fixed at 128.

three metrics: AUROC, AUPRC, and validation loss (MSE loss). Table 6 summarizes the results. As expected, the exact computation yields optimal performance. Regarding the approximations, we observe an improvement in accuracy as $R$ increases, progressively closing the performance gap with the exact estimator. Crucially, the absence of regularization leads to a significant degradation in AUPRC, indicative of a high false positive rate. This empirically validates the necessity of regularization for suppressing spurious edges to ensure robust causal discovery.

*Table 6.* Sensitivity to the number of random projections evaluated on the first five subjects of the fMRI dataset. Metrics include AUROC, AUPRC, and validation loss (prediction error). Best results are highlighted in **bold**.

| Penalty | AUROC | AUPRC | Val loss |
|---|---|---|---|
| None | 0.671 (0.010) | 0.328 (0.040) | 1.060 (0.038) |
| F-Shap (1) | 0.848 (0.022) | 0.634 (0.036) | 0.964 (0.032) |
| F-Shap (3) | 0.848 (0.028) | 0.634 (0.033) | 0.963 (0.038) |
| F-Shap (5) | 0.851 (0.022) | 0.638 (0.042) | 0.961 (0.034) |
| Shap | **0.856** (0.022) | **0.676** (0.019) | **0.946** (0.038) |

## D.4. Additional Experiments Results

Figure 4 visualizes the Info-Shap importance scores from the experiments in Section 4.3, demonstrating the validity of our method. Specifically, the original features (upper panel) faithfully represent the underlying causal structure. A detailed examination reveals that diagonal entries exhibit a magnitude attenuation consistent with the decay of auto-correlation over lags; crucially, entries at lag 4 (where no true causality exists) drop to negligible levels, verifying that Info-Shap accurately measures the importance of features. In contrast, the knockoff features (lower panel) receive uniformly suppressed scores, confirming the efficacy of the knockoff thresholding in distinguishing predictive signals from noise.

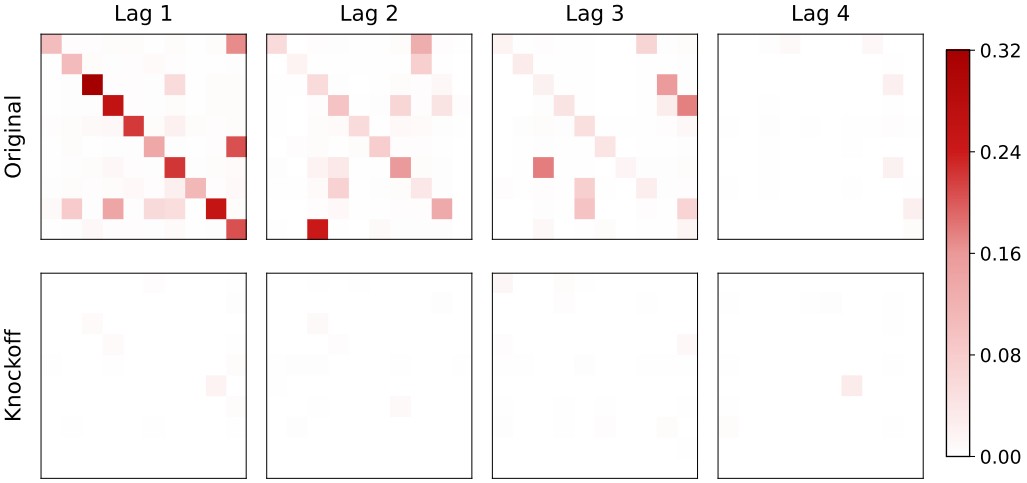

*Figure 4.* **Visualization of Info-Shap importance scores.** Heatmaps depicting importance scores for original features (upper panel) and knockoff features (lower panel), derived from an SRNGC model trained on a VAR(3) dataset with a lag order of 4. Darker colors indicate higher importance score.

