# OpenReview forum: "Shapley Regularized Neural Granger Causality"
_ICML.cc/2026/Conference — ICML 2026 regular_

### Official Review · Reviewer_vTMe · 2026-02-20

**Soundness:** 4
**Presentation:** 3
**Significance:** 4
**Originality:** 4
**Overall Recommendation:** 6
**Confidence:** 4

**Summary:**

The authors touch upon an interesting dimension in neural Granger causality, where the previous works would focus on the complex neural architectures which were capable of capturing non-linear dynamics, but fail to identify them.

Therefore, they introduce a
=> new measure, which is the importance measure, where they prove that by understanding the information contribution of a specific series to another (ith to jth) one can interpret the granger causality.
=> This is then used to introduce shapely regularizer, which aims to use the previous finding, into decreasing the effects of dense graphs, which often contradict real-world systems. This is then approximated using Frobenius norm to introduce F-shap. A knockoff threshold is then introduced, which aims to decrease the effects noise and approximation errors.
=> Residual MLP is leveraged to model the underlying dynamics

For the exps, the authors experienced with both synthetic and realistic datasets. They then test the knockoff threshold effects

Their findings, confirm that while F-shap can introduce a slight approximation degradation, especially for small sample size such as Dream3, with the right sample size it can approximate Shap and even exceed baselines. Furthermore, knockoff threshold proved to be able to rebuilding the ground truth without much of FPs and TPs.

**Compliance With Llm Reviewing Policy:**

Affirmed.

**Key Questions For Authors:**

if the users may at least compare with the case of not using the threshold, as this would further justify why to use the knockoff (this is the most important factor)

while the other 2 points are somewhat optional, but they would definitely help further justify the claims
- compare against a baseline model, while this is has been left for future work, if it is possible, this would definetly support the accuracy drop for the medicine dataset

- a simple measurement of how much F-shap saves on time, would justify the accuracy drop.

**Limitations:**

yes

**Strengths And Weaknesses:**

The reviewer likes the clarity of the 2 new added losses, and how that even that F-shap being an approximation is capable of exceeding the baselines, while as well being comparably similar if not outright exceeding shap for the correct sample size.

The idea of that interpreting granger causality using marginal info contribution is interesting, and how it was implemented through the regularizer is well clarified.

The generality of the model, being able to be adopted to any temporal dynamic model is especially useful for further research.

The use of both a controlled synthetic dataset, and a real-work dataset further highlight the merits of using F-shap.

Some comments to further enhance the work would be
- while both Shap and F-shap performs well for most of the datasets in Causal time dataset, claiming that the reason for underperforming on medicine dataset was the underlying model needs further justification. While it is understandable that this might be the reason   for underperforming, the work would definitely benefit from testing the same losses on other (at least another temporal model). This will not only further justify the results, but it will further support the claims that the used losses can be further applied to multiple models.

- in the knockoff threshold experiment, while the results are convincing, where the use of knockoff thresholding procedure was capable of rebuilding the whole ground truth with limited FNs and FPs, this needs to be compared with the case of not using the knockoff threshold. This would further support the merits of using this procedure, the authors, if possible might as well compare this threshold technique with other basic methods, to further support their claims.


- additionally, another simple measurement of how F-shap saves on running time, may as well further justify the slight decrease in accuracy.

---

> ### Author Rebuttal · Authors · 2026-03-31
>
> We sincerely thank Reviewer vTMe for the constructive comments and positive feedback. Your insights have been invaluable in guiding our efforts to improve our work. We address each of your key questions below.
>
> **Q1: If the users may at least compare with the case of not using the threshold, as this would further justify why to use the knockoff (this is the most important factor).**
>
> **A1:** To demonstrate the necessity of the knockoff procedure, we evaluated it against a "no-threshold" baseline and naive Top-$K$ heuristics ($K=20, 30, 40$). The resulting F1-score, AUROC, and AUPRC for each method are reported below:
>
> | Threshold Method | F1-score | AUROC | AUPRC |
> | :---: | :---: | :---: | :---: |
> | Knockoff | 1.00 | 1.00 | 1.00 |
> | Top-20 | 0.80 | 0.83 | 0.77 |
> | Top-30 | 1.00 | 1.00 | 1.00 |
> | Top-40 | 0.86 | 0.93 | 0.75 |
> | Non-threshold | 0.46 | 0.50 | 0.30 |
>
> We summarize the key observations as follows:
> * A thresholding step is necessary in practice. Due to finite-sample noise, estimated importance scores are rarely exactly zero; without a proper threshold, these small non-zero values introduce massive false positives, drastically degrading the F1-score to 0.46.
> * Both the Knockoff and Top-$30$ faithfully recover the true causal structure, demonstrating that the proposed Info-Shap correctly ranks feature importance.
> * The Top-$30$ heuristic achieves a perfect F1-score only because it acts as an oracle, relying on the fact that the ground-truth sparsity level in this specific setting is known to be exactly 0.3. In practice, where such prior knowledge is unavailable, the performance of Top-$K$ strategies degrades significantly if $K$ is misspecified. In contrast, our proposed knockoff procedure is fully data-driven, achieving oracle-level performance without requiring any prior information regarding the underlying sparsity.
>
> **Q2: Compare against a baseline model, while this is has been left for future work, if it is possible, this would definitely support the accuracy drop for the medicine dataset.**
>
> **A2:** To further investigate the impact of the underlying predictive model and strengthen our empirical claims, we evaluated our method across three additional backbone architectures. Specifically, we incorporated a modern architecture (Informer [1]) alongside two standard baselines (MLP and SRU [2]). We focused this extended evaluation on the Medical dataset using the F-Shap penalty. To ensure a fair and rigorous comparison, the entire training pipeline was kept strictly identical to the setup described in the main text. The results are summarized below:
>
> | Metric | MLP | SRU | Informer  |
> | :---: | :---: | :---: | :---: |
> | AUROC | 0.760 | 0.789 | 0.712 |
> | AUPRC | 0.705 | 0.776 | 0.646 |
>
> Empirical evaluations demonstrate the broad applicability of our proposed method across diverse architectures, including Transformer-based models like the Informer. Furthermore, the method achieves competitive performance when integrated with standard architectures such as MLP and SRU. These findings support our view that the performance drop on the Medical dataset is due to the limitations of the predictive model, confirming that the proposed methodology remains robust and effective.
>
> **Q3: A simple measurement of how much F-Shap saves on time, would justify the accuracy drop.**
>
> **A3:** To provide deeper insight into the efficiency-accuracy trade-off, we compared the computational costs of exact SRNGC and Fast-SRNGC (F-SRNGC). All experiments were conducted on a server equipped with an NVIDIA A800 GPU. The average per-epoch runtime and total execution time across the DREAM3 datasets are reported below:
>
> | Method | Metric | Ecoli-1 | Ecoli-2 | Yeast-1 | Yeast-2 | Yeast-3 |
> | :--- | :--- | :---: | :---: | :---: | :---: | :---: |
> | SRNGC | AUROC | 0.700 | 0.702 | 0.665 | 0.611 | 0.596 |
> | | Epoch Time (s) | 0.21 | 0.23 | 0.13 | 0.49 | 0.83 |
> | | Total Time (s) | 756.25 | 518.40 | 27.18 | 408.97 | 119.82 |
> | F-SRNGC | AUROC | 0.632 | 0.642 | 0.665 | 0.604 | 0.611 |
> | | Epoch Time (s) | 0.06 | 0.05 | 0.06 | 0.08 | 0.08 |
> | | Total Time (s) | 9.69 | 20.15 | 32.13 | 45.60 | 9.92 |
>
> As the results demonstrate, F-SRNGC is generally faster in both average per-epoch and total running time. Furthermore, it achieves performance comparable to SRNGC on the Yeast dataset, supporting F-SRNGC as a practical and scalable alternative when the computational cost of SRNGC is prohibitively expensive.
>
> **We hope our responses have successfully addressed your concerns.**
>
> **References**
>
> [1] Zhou, Haoyi, et al. "Informer: Beyond efficient transformer for long sequence time-series forecasting." Proceedings of the AAAI conference on artificial intelligence. Vol. 35. No. 12. 2021.
>
> [2] Oliva, Junier B., Barnabás Póczos, and Jeff Schneider. "The statistical recurrent unit." International Conference on Machine Learning. PMLR, 2017.

---

> > ### Author Rebuttal · Reviewer_vTMe · 2026-03-31
> >
> > Thanks for your response, my questions were fully answered (and the score was raised accordingly in soundness from 3->4), and from (accept -> strong accept)
> > Just a couple of points you might consider when including these results in the paper
> >
> > * for the Top-K experiment, you need to highlight the factors that would make oracle behavior degrade with higher top-k, if it is dataset specific, you might need to highlight this, and if the space allows, you might include the same experiment onto different datasets.
> >
> > * for the exp of the underlying exp, you might need to highlight the correlations of the underlying architecture with the dataset being utilized. Specifically, include a reasoning to when AUPRC might inc/dec from going to linear architecture to a transformer architecture. This would allow the community to know when to use specific architectures as the underlying predictive models.
> >
> > Overall, the idea is well articulated, and discuses an essential aspect of causal modeling.

---

> > > ### Author Response · Authors · 2026-04-05
> > >
> > > **We sincerely thank the reviewer for the highly encouraging feedback and thoughtful follow-up. We are very pleased that our rebuttal has fully addressed your concerns, and we greatly appreciate your stronger endorsement of the paper. We also thank you for the valuable suggestions, which we will carefully consider and incorporate as much as possible in the final manuscript.**

---

### Official Review · Reviewer_LQa5 · 2026-03-11

**Soundness:** 3
**Presentation:** 3
**Significance:** 3
**Originality:** 3
**Overall Recommendation:** 4
**Confidence:** 3

**Summary:**

This paper addresses the misalignment between inference mechanisms and model expressive power in Neural Granger Causality by proposing Info-Shap—an information-theoretic Shapley value based on a conditionally generated information matrix. Under the Gaussian assumption, it possesses a closed-form expression and is equivalent to Granger noncausality theory. Based on this, two model-independent regularizers, Shap and F-Shap, are designed and can be integrated into the training of any differentiable neural network, outperforming existing Jacobian baselines on multiple synthetic and real datasets.

**Compliance With Llm Reviewing Policy:**

Affirmed.

**Final Justification:**

The authors' rebuttal addressed my main concerns. I have accordingly raised my recommendation.

**Key Questions For Authors:**

1. Does Info-Shap still have a closed-form expression under non-Gaussian noise? Does Theorem 3.2 still hold?

2. Can you provide specific counterexamples to illustrate that adding Info-Shap in 2-optional approaches misses important causal relationships? Under what conditions is the second-order approximation sufficient?

3. What is the theoretical relationship between the solution spaces induced by F-Shap and the exact Shap? Under what conditions are the causal discovery results consistent?

4. When the predictive model deviates significantly from the true function, how does the causal discovery guarantee of Info-Shap degrade?

5. Do the Knockoff features constructed by time series residual permutations satisfy the exchangeability condition, and is FDR control still effective?

**Limitations:**

yes

**Strengths And Weaknesses:**

Strengths:

The framework unifying NGC methods into a triplet of importance metric, prediction model, and penalty term is insightful and convincingly demonstrates the limitations of local importance metrics.

Info-Shap establishes formal equivalence to Granger causality through the virtual player axiom. Its 2-additive structure reduces the exact computation from O(2^N) to O(N^2), resulting in an elegant theoretical construction.

The regularizer depends only on Jacobian and does not constrain the network architecture. Model independence allows the same inference mechanism to be applied to different architectures such as MLP and LSTM.

Weaknesses:

Info-Shap's closed-form expression and core theorems rely entirely on the Gaussian assumption. The paper claims it can be generalized to non-Gaussian distributions but provides no formulas or experimental verification.

The coalition value function has a 2-additive capacity. Info-Shap only captures pairwise interactions while completely ignoring higher-order effects, creating tension with the paper's claim of "capturing complex nonlinear dependencies."

F-Shap replaces the $\ell_1$ norm with the squared Frobenius norm, which is a different regularization objective rather than an approximation. It lacks a theoretical error bound, and Table 5 shows that the AUPRC difference is not negligible (0.676 vs 0.638).

In the VAR(3) experiment, the four methods performed almost identically, failing to demonstrate any nonlinear advantage. On the Medical dataset, they performed worse than LCCM, revealing a strong dependence of the methods on the quality of the prediction model.

Knockoff features are constructed by permuting residuals, but it is not proven that they satisfy the exchangeability condition under the time series autocorrelation structure, and the computational cost doubles.

---

> ### Author Rebuttal · Authors · 2026-03-31
>
> We sincerely thank the reviewer for your detailed and technically substantive review.  Our point-by-point responses are below.
>
> **Q1. Does Info-Shap still have a closed-form expression under non-Gaussian noise? Does Theorem 3.2 still hold?**
>
> **A1.**  The closed-form expression of Info-Shap relies fundamentally on the 2-additive capacity of the value function. In non-Gaussian settings, treating the score function as a function of the inputs allows for the formulation of a generalized value function that preserves this property. Consequently, Theorem 3.2 could be strengthened to imply conditional independence.
>
> The Gaussian assumption was adopted primarily for analytical tractability and to present our core ideas clearly. While extending the framework to non-Gaussian distributions is conceptually feasible, we defer its rigorous theoretical development and empirical validation to future work.
>
> **Q2.Can you provide specific counterexamples to illustrate that adding Info-Shap in 2-optional approaches misses important causal relationships? Under what conditions is the second-order approximation sufficient?**
>
> **A2.** While our value function utilizes a 2-additive capacity, this formulation does not ignore higher-order effects; instead, such dependencies are implicitly captured by Info-Shap. To demonstrate this, consider a Data Generating Process defined by a third-order interaction: $y = x_1x_2x_3+\epsilon$. Calculating the Info-Shap for feature 1 yields:
>     $$\phi_1=E[|X_2^2 X_3^2|+0.5|X_1X_2X_3^2|+0.5|X_1X_3X_2^2|],$$
> which is generally positive. Within Info-Shap, the underlying Shapley axioms ensure that higher-order dependencies are reflected in the resulting attributions.
>
> **Q3. What is the theoretical relationship between the solution spaces induced by F-Shap and the exact Shap? Under what conditions are the causal discovery results consistent?**
>
> **A3.** Both Shap and F-Shap regularize individual and interaction effects. The key difference lies in how the interaction terms are penalized: Shap induces a sparsity-promoting $\ell_1$-type structure, while F-Shap adopts a squared Frobenius formulation, yielding a smoother optimization landscape. Generally speaking, both regularization schemes aim to encourage the gradient of the estimated function to approach zero.
>
> Since the penalty is applied to the gradients of a nonlinear neural network evaluated over the data distribution, the resulting optimization problem involves complex interactions between the model, the data, and the attribution structure. This makes a rigorous theoretical analysis of the induced solution space challenging. We therefore leave a detailed comparison between Shap and F-Shap, as well as consistency analysis of the resulting causal graphs, for future work.
>
> **Q4. When the predictive model deviates significantly from the true function, how does the causal discovery guarantee of Info-Shap degrade?**
>
> **A4.** We agree that, as a model-based framework, our method's efficacy relies on the quality of the underlying predictive model. When the model does not adequately approximate the true regression function, the resulting performance naturally deteriorates, as observed in the Medical dataset.
>
> For VAR(3), where the underlying dynamic is linear, the baseline Jacobian method can capture the structure effectively; thus, achieving comparable performance is expected (Table 1). The advantages of our framework become more pronounced in nonlinear settings, such as the Lorenz-96 and other realistic benchmarks (Table 2, 3 and 4).
>
> Therefore, our main contribution is not a new predictive backbone, but rather a model-agnostic methodology for temporal causal discovery. Specifically, the proposed Info-Shap measure and Shapley regularizers establish a faithful attribution mechanism, which empirically achieves consistent performance improvements across diverse predictive architectures.
>
> **Q5. Do the Knockoff features constructed by time series residual permutations satisfy the exchangeability condition, and is FDR control still effective?**
>
> **A5.** The answer to both questions is no. The knockoff features from time-series residual permutations do not satisfy the exchangeability, mainly because of the serial dependence inherent in time series. Hence, our method should not be interpreted as providing a formal FDR control guarantee. Instead, our “knockoff-style” step serves as a practical, data-driven thresholding procedure that empirically helps suppress false discoveries; see Section 4.3 and Appendix D.4.
>
> While fitting the augmented model adds computational overhead, it is strictly a post-processing step for final graph selection, not required for main training or ranking metrics like AUROC. In the revision, we will clarify this empirical scope and develop time-series knockoff theory in future work.
>
> **Due to space limits, we can address the rest during next phase. We hope our responses and planned revisions have successfully addressed your concerns**

---

> > ### Author Rebuttal · Reviewer_LQa5 · 2026-03-31
> >
> > I thank the authors for their reply. I am satisfied with the response, and have no further comments.

---

> > > ### Author Response · Authors · 2026-04-05
> > >
> > > **We sincerely thank the reviewer for carefully considering our rebuttal and for the thoughtful feedback throughout the review process. We are pleased that our point-by-point responses have addressed your concerns and helped clarify the scope and contributions of our work. We greatly appreciate your updated positive assessment.**

---

### Official Review · Reviewer_KYj8 · 2026-03-12

**Soundness:** 3
**Presentation:** 4
**Significance:** 3
**Originality:** 3
**Overall Recommendation:** 5
**Confidence:** 3

**Summary:**

The authors present a method to find the Granger causality between time series data by formulating the problem as a feature attribution problem. They find statistical relationships between features in time series data, using sharply values to determine whether one feature may be predictive of another. They propose a new formulation of the Shapley value suited to neural grander causality models they call Information-Theoretical Shapley value and show that this value is zero if and only if a target Y is independent on other features given that particular feature. They show in empirical results that their method is successful in uncovering relationships between graphs and outperforms benchmarks in a large proportion of settings investigated.

**Compliance With Llm Reviewing Policy:**

Affirmed.

**Key Questions For Authors:**

My key question to the authors is: outside of the performance gains this method could potentially provide, what are the other contributions that this method make? Could the unique formulation of the problem result in increased interpretability of the system / perhaps potential to uncover the weights of the graph, rather than just the existence of edges etc.

**Limitations:**

Yes

**Strengths And Weaknesses:**

Soundness:
The claims made in this paper are largely well supported by evidence. The method designed seems like a reasonable approach for solving this problem. The only things that I was unsure about was why they do not show the performance comparison on the VAR (3) dataset and the Lorenz-96 dataset with all the methods which are used on the DREAM3 and CausalTime datasets in Table 3 and 4? For the VAR (3) dataset and the Lorenz-96 dataset, they only compare performance of different regularisation methods, jacob-l1 etc, rather than the different temporal causal discovery approaches as done on the DREAM3 and CausalTime datasets.
Are these two categories of datasets extremely different? Why are there no results on the temporal causal discovery approaches for the VAR (3) dataset and the Lorenz-96 datasets.

Presentation:
The presentation of the paper is very good and, therefore easy to understand. There was a section in the paper, section 3.2 where they use sub-scripts to denote sub-sets of the features i, which confused me for a moment, since I believe that in the rest of the paper, super-scripts are reserved for features and sub-scripts are reserved for time t. So if I am not mistaken in this, the clarify of the paper could be improved by sticking to sub-scripts only for time t and super-scripts only for features.

Significance:
The contribution of the paper seems significant, since it outperforms other baselines on benchmark tasks.

Originality:
The formulation of this problem as a feature attribution problem using shapely values is novel and interesting.

---

> ### Author Rebuttal · Authors · 2026-03-31
>
> We are grateful for the positive reception of our work and the constructive feedback provided. We appreciate the time dedicated to reviewing our manuscript. Below, we address your specific questions regarding the experimental design, notational consistency, and the broader contributions of our framework.
>
> **Q1: The only things that I was unsure about was why they do not show the performance comparison on the VAR (3) dataset and the Lorenz-96 dataset with all the methods which are used on the DREAM3 and CausalTime datasets in Table 3 and 4?**
>
> **A1:** The synthetic datasets (VAR(3) and Lorenz-96) and the realistic datasets (DREAM3 and CausalTime) are served different evaluative purposes:
> * For the synthetic datasets, our goal is to perform *an ablation study*. Since the true lag order and the functional form of the causal relationships are known in these settings (linear for VAR(3) and nonlinear for Lorenz-96), we can fix the predictive backbone to isolate the impact from the importance measure and regularizer. Under this design, performance differences can be attributed directly to the attribution mechanism itself, rather than to differences in model architecture, optimization strategy, or other components of a full causal discovery pipeline. As shown in Section 4.1, this controlled comparison reveals that our method performs comparably to the Jacobian-based approach in the linear VAR setting (Table 1), while showing clearer advantages in the nonlinear Lorenz-96 setting (Table 2).
>
> * By contrast, DREAM3 and CausalTime are used for *end-to-end benchmark evaluation*. In Section 4.2, we therefore compare against a broader set of temporal causal discovery methods on these datasets (Table 3 and 4), where the goal is to assess overall practical performance rather than to isolate the role of a single component.
>
> We truly appreciate your feedback and will explicitly clarify this experimental rationale in the revised manuscript to strengthen the paper's presentation.
>
> **Q2: The clarify of the paper could be improved by sticking to sub-scripts only for time t and super-scripts only for features.**
>
> **A2:** We agree that the presentation can be improved, and we will revise the manuscript to ensure consistent notation throughout the paper. Specifically, we will strictly reserve subscripts for time $t$ and superscripts for features. For example, the expression for Info-Shap will be updated from $\phi_i(\cdot)$ to $\phi^i(\cdot)$.
>
> **Q3: Outside of the performance gains this method could potentially provide, what are the other contributions that this method make? Could the unique formulation of the problem result in increased interpretability of the system / perhaps potential to uncover the weights of the graph, rather than just the existence of edges etc.**
>
> **A3:** We sincerely thank for raising this insightful comment, as it highlights the broader implications of our work. Beyond empirical performance gains, our formulation provides a principled way to quantify the strength of relationships in the graph and enhances the interpretability of the system from an information-theoretic perspective.
>
> Our primary methodological contribution is presenting an alternative perspective on temporal causal discovery with deep learning by formulating it as a feature attribution problem. To operationalize this concept, we introduce Info-Shap, an information-theoretic measure grounded in the Shapley value framework. As Info-Shap quantifies the information contribution of the features in predicting the response, it can be naturally extended to assess the magnitude of these contributions.
>
> Furthermore, Info-Shap can be applied in a post-hoc manner to analyze the importance of individual features, groups of features, or specific components within the network architecture. This capability illuminates how information is distributed and propagated throughout the model, thereby providing interpretable insights into the underlying system dynamics.
>
> ***We hope our responses and planned revisions have successfully addressed your concerns.***

---

> > ### Author Rebuttal · Reviewer_KYj8 · 2026-04-03
> >
> > I will maintain my recommendation to accept this paper. The authors have addressed my remaining questions sufficiently with their response.

---

> > > ### Author Response · Authors · 2026-04-05
> > >
> > > **We sincerely thank the reviewer for their thoughtful evaluation and encouraging feedback. We are glad that our rebuttal has sufficiently addressed the remaining questions, and we greatly appreciate the reviewer’s recommendation to accept the paper.**

---

### Decision · Program_Chairs · 2026-04-30

**Decision:**

Accept (regular)

**Comment:**

The paper reformulates neural Granger causality as a feature attribution problem and proposes Info-Shap, an information-theoretic Shapley value with proven equivalence to Granger non-causality under Gaussian assumptions. Two model-agnostic regularizers (Shap and F-Shap) and a knockoff-style thresholding procedure are introduced. Reviewers agreed that the contribution is original and well-executed:

1. **Originality.** The information-theoretic Shapley formulation is novel, the 2-additive capacity yields O(N^2) computation, and the unifying framework for existing NGC methods is insightful (`LQa5`). The model-agnostic design allows integration into any differentiable architecture.

2. **Soundness.** The core theory (Theorems 3.1-3.2) is elegant and well-constructed. The rebuttal substantially strengthened the empirical case with three new experiments: knockoff vs. no-threshold/Top-K comparison (demonstrating necessity of the thresholding step), multi-architecture evaluation across MLP, SRU, and Informer (`vTMe`), and detailed runtime analysis showing F-Shap as a practical scalable alternative.

3. **Empirical results.** Consistent improvements over Jacobian-based baselines across four benchmark suites (VAR(3), Lorenz-96, DREAM3, CausalTime). `vTMe` raised their score to strong accept after the rebuttal.

Scores were 6, 4, 5 (mean 5.0). All three reviewers marked concerns as fully resolved. `vTMe` raised from accept to strong accept; `LQa5` raised their recommendation after the rebuttal. The paper is recommended for acceptance.

**Revision requirements.** The gap between the Gaussian theory and the practical methods should be clearly acknowledged in the camera-ready:
- F-Shap uses the squared Frobenius norm, which is a different regularization objective from the l1-based Shap, not an approximation. The paper should state this explicitly.
- The knockoff procedure does not satisfy exchangeability under time-series autocorrelation and provides no formal FDR guarantee. The authors acknowledged this during rebuttal but this limitation should be stated in the main text.
- The claim that Info-Shap "can be generalized to non-Gaussian distributions" should be softened until supported by theory or experiments.